# MEMORY-CONSISTENT NEURAL NETWORKS FOR IMITATION LEARNING

**Kaustubh Sridhar**[1], **Souradeep Dutta**[1], **Dinesh Jayaraman**[1], **James Weimer**[1,2], **Insup Lee**[1]
[1]University of Pennsylvania, [2]Vanderbilt University
`{ksridhar, duttaso, dineshj, weimerj, lee}@seas.upenn.edu`

## ABSTRACT

Imitation learning considerably simplifies policy synthesis compared to alternative approaches by exploiting access to expert demonstrations. For such imitation policies, errors away from the training samples are particularly critical. Even rare slip-ups in the policy action outputs can compound quickly over time, since they lead to unfamiliar future states where the policy is still more likely to err, eventually causing task failures. We revisit simple supervised "behavior cloning" for conveniently training the policy from nothing more than pre-recorded demonstrations, but carefully design the model class to counter the compounding error phenomenon. Our "memory-consistent neural network" (MCNN) outputs are hard-constrained to stay within clearly specified permissible regions anchored to prototypical "memory" training samples. We provide a guaranteed upper bound for the sub-optimality gap induced by MCNN policies. Using MCNNs on 10 imitation learning tasks, with MLP, Transformer, and Diffusion backbones, spanning dexterous robotic manipulation and driving, proprioceptive inputs and visual inputs, and varying sizes and types of demonstration data, we find large and consistent gains in performance, validating that MCNNs are better-suited than vanilla deep neural networks for imitation learning applications.[1]

## 1 INTRODUCTION

For sequential decision making problems such as robotic control, imitation learning is an attractive and scalable option for learning decision making policies when expert demonstrations are available as a task specification. Such demonstrations are typically easier to provide than the typical task specification requirements for reinforcement learning and model-based control, namely, dense rewards and good models of the environment. Furthermore, imitation learning is also typically less experience-intensive than reinforcement learning and less expertise-intensive than model-based control.

We consider the simplest and perhaps most widely used imitation learning algorithm, behavior cloning (BC) [29], which reduces policy synthesis to supervised learning over the expert demonstration data. For example, a neural network policy for an autonomous car could be trained to mimic human driving actions [2]. While the policy is synthesized with supervised learning, the evaluation setup is very different: rather than merely achieving low average error on states from the training data, as common in supervised learning, the trained policy must, when rolled out in the world, successfully accomplish the demonstrated task.

This sequential deployment makes the behavior of imitation policy functions away from their training data particularly critical. To see this, observe that during rollout, the policy's own output actions determine its future input states. Task performance is most closely tied to the policy's behavior on this self-induced set of states, which can deviate from the training dataset of expert demonstrations. In particular, a minor error in the policy's action output at any time may induce a future input state that is subtly different from expert states. If the policy behaves erratically under such small deviations, as it often does in practice, the situation quickly snowballs into a vicious cycle of compounding errors leading to task failure.

---

[1] Our website: https://sites.google.com/view/mcnn-imitation.

Past solutions to this compounding error problem have focused on modifying the behavior cloning setup, such as by permitting online experience [13, 34], reward labels [28], queryable experts [37], or modifying the demonstration data collection procedure [19]. Instead, we retain the conveniences of the plain BC setup and focus on designing a model class that encourages better behavior beyond the training data, which in turn could boost task performance by mitigating the compounding error phenomenon discussed above. We provide a simple plug-in approach to improve BC with any deep neural network.

It is well known that vanilla deep neural networks, only by themselves, can generate large errors when evaluated away from the training points, and even rare errors could derail an entire task rollout. These large errors are particularly evident when the expert demonstrations are few in number such as in robotics where human demonstrations are essential for imitation learning. To tame these errors, we propose semi-parametric "memory-consistent neural networks" (MCNN). MCNNs first subsample the dataset into representative prototype "memories" to form the scaffold for the eventual function. They then fit a parametric function to the rest of the training data that is hard-constrained by the very formulation of the model class, to exactly fit the training data at all the memories, and further, to stay within double-cone-like zones of controllable shapes and sizes centered at each memory. As a result, an MCNN behaves mostly like a nearest-neighbor function close to memories, and mostly like a deep neural network (subject to the double-cone constraints) far from them. All functions in this MCNN model class lie within "permissible regions" centered on each memory, meaning that function values away from the training points are bounded. Under mild assumptions on the expert policy, we show that this property of MCNNs induces an upper bound on the suboptimality of the learned BC policy. Visualizations of MCNNs can be found in Figure 2.

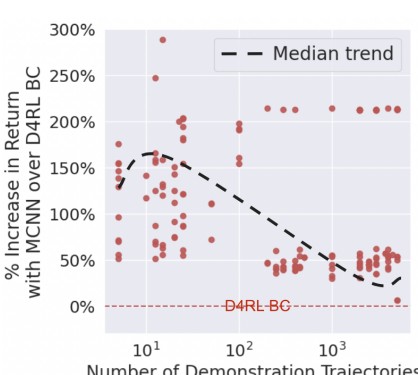

Figure 1: **MCNN significantly improves performance on realistic demonstration datasets.** We plot the percentage increase in return with MCNN over D4RL BC [9] for various number of demonstrations across many tasks. In this plot, each point is a separate MCNN policy. We see significant improvements in the few demonstrations regime where most realistic imitation learning tasks can be found. The choice of model class is crucial in such regimes and MCNN shines. Additional details are in Appendix F.

Using MCNNs on 10 imitation learning tasks, with MLP, Transformer and Diffusion backbones, spanning dexterous robotic manipulation and driving, proprioceptive inputs and visual inputs, and varying sizes and types of demonstration data, we find large and consistent gains in performance, validating that MCNNs are better-suited than vanilla deep neural networks for imitation learning applications. Figure 1 visualizes the percentage increase in return with MCNN policies compared to the vanilla BC results reported in D4RL [9] for various quantities of training demonstrations across tasks. The trend of the median demonstrates that MCNNs are highly effective in the low data regime where generalization to test trajectories is stressed.

## 2 RELATED WORK

We present a detailed related work discussion in Appendix D & summarize closely related work here.

**Compounding errors in imitation learning** have previously been tackled by permitting online experience [13, 34], reward labels [28], queryable experts [37], or modifying the demonstration data collection procedure [19]. Our work is orthogonal to these methods and creates a model class that avoids compounding errors by construction. Other works that propose new models for IL such as Implicit BC (IBC) [7], Behavior Transformer (BeT) [38], Action Chunking Transformer [48], and Diffusion Policies [44, 1] are orthogonal to our approach. MCNN can be used as a plug-in approach to improve any of these methods. In fact, we show that MCNN with a BeT backbone outperforms vanilla BeT and MCNN with a diffusion model outperforms diffusion BC on all tasks in our experiments in Section 5. We also show that MCNN outperforms IBC in Section 5.

**Non-parametric and semi-parametric methods in imitation learning** such as nearest neighbors [39], RBFs [35], and SVMs [20] have historically shown competitive performance on various robotic control benchmarks. But, only recently, a semi-parametric approach consisting of neural networks for representation learning and k-nearest neighbors for control was proposed in Visual Imitation through

Nearest Neighbors (VINN) [26]. This is the closest paper to our work and in Section 5, we compare with VINN and demonstrate that we outperform their method comprehensively.

**Theoretical guarantees on the sub-optimality gap in imitation learning** with MCNN are provided in this paper. Such guarantees are not available with vanilla neural networks. Our theorem builds on earlier work on reductions for imitation learning in [36, 31, 2, 32] and leverages intuitions from [25] on bounding the width of the model class.

## 3 PROBLEM FORMULATION

A *Markov Decision Process (MDP)* is a tuple $\mathcal{E} = (\mathcal{S}, \mathcal{A}, \mathcal{P}, \mathcal{R}, \gamma, \mathcal{I})$, where $\mathcal{S} \subseteq \mathbb{R}^n$ is the set of states, $\mathcal{A}$ is the set of actions, $\mathcal{P}(s'|s, a)$ is the probability of transitioning from state $s$ to $s'$ when taking action $a$, $\mathcal{R}(s, a)$ is the reward accrued in state $s$ upon taking action $a$, $\gamma \in [0, 1)$ is the discount factor, and $\mathcal{I}$ is the initial state distribution. We assume that the MDP operates over trajectories with finite length $H$, in an episodic fashion. Additionally, we assume the set of states $\mathcal{S}$ to be closed and compact. Given a policy $\pi : \mathcal{S} \mapsto \mathcal{A}$, the expected cumulative reward accrued over the duration of an episode is given by the following,

$$J(\pi) = \mathbb{E}_\pi \Big[ \sum_{t=1}^{H} \mathcal{R}(s_t, a_t) \Big]. \tag{1}$$

In imitation learning, we assume that there exists an expert policy $\pi^*$ unknown to the learner. This policy induces a distribution $d_{\pi^*}$ on the state-action space $\mathcal{S} \times \mathcal{A}$ obtained by rollouts on the MDP. The learner agent has access to an expert trajectory dataset $D = \big\{(s_0, a_0), (s_1, a_1), \ldots, (s_N, a_N)\big\}$ drawn from distribution $d_{\pi^*}$. The goal of imitation learning is to estimate a policy $\hat{\pi}$, which mimics the expert's policy and reduces the *sub-optimality* gap: $J(\pi^*) - J(\hat{\pi})$.

## 4 APPROACH

Our approach involves developing a new model class, memory consistent neural networks (MCNN), and training it with supervised learning to clone the expert from the demonstration data. We start by setting up the MCNN model class in Sec 4.1, analyze its theoretical properties for imitation learning in Sec 4.2, and finally describe our behavior cloning algorithm that uses MCNNs in Sec 4.3.

### 4.1 THE MODEL CLASS: MEMORY-CONSISTENT NEURAL NETWORKS

First, we develop the semi-parametric MCNN model class for imitation learning. MCNNs rely on a code-book set of "memories" $\mathcal{B} := \{(s_i, a_i)\}_{i=1}^{K}$ which are subsampled from the expert training dataset and summarize it. In practice, such a memory code-book can be created using one of various off-the-shelf approaches. We describe our algorithmic choices later in Sec 4.3. For notational convenience, we describe the approach for a scalar action space, but it is easily generalizable to the vector action spaces we evaluate in our experiments.

Given this memory code-book $\mathcal{B}$, we now define a "nearest memory neighbor policy". For a finite set $S \subset \mathcal{S}$, and an input $x \in \mathcal{S}$, we first define its closest element in $S$ as, $C_S(x) = \arg\min_{s \in S} d(s, x)$,
where $d$ is some distance metric defined on the space $\mathcal{S}$. We denote by $\mathcal{B}|_S$, and $\mathcal{B}|_A$ as the set of all states and actions captured by the memory code-book $\mathcal{B}$. With slight abuse of notation, we denote $\mathcal{B}(s)$ as the action assigned by the codebook for a state input $s$. Using the above, we now define a nearest neighbor regression function $f^{NN}$ as the following,

**Definition 4.1** (Nearest Memory Neighbor Function). For an input $x \in \mathcal{S}$, assume that $s' = C_{\mathcal{B}|_S}(x)$, then $f^{NN}(x) := \mathcal{B}(s')$.

In other words, the nearest memory neighbor function assigns actions according to a nearest neighbor look-up in the memory code-book $\mathcal{B}$. We are now ready to define *memory-consistent neural networks* (MCNN), which permit interpolating between nearest neighbor functions and parametric deep neural network (DNN)-based functions. Let $f^\theta$ denote a DNN function parameterized by $\theta$, which maps from MDP states to actions.

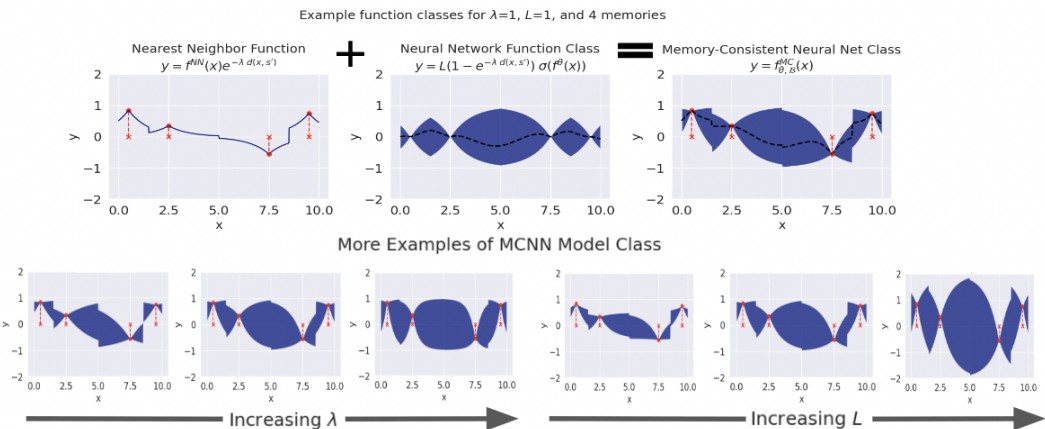

Figure 2: **The elements of the MCNN model class.** In the top row, the left panel shows the nearest memory neighbour component with memories subsampled from the training dataset shown in red circles. The middle panel depicts the constrained neural network function class, where the blue shaded regions represent the permissible regions; by design, the function cannot take values outside these shaded regions. Finally, the right panel shows the combined MCNN model class. The size of the permissible regions can be modulated by increasing $\lambda$ (bottom left) or by decreasing the number of memories (bottom right). The second row shows many such MCNN model families with increasing capacity. For additional plots, see Appendix A.

**Definition 4.2** (Memory-Consistent Neural Network). A memory-consistent neural network $f^{MC}$ is defined using the codebook and DNN function pair $(\mathcal{B}, f^\theta)$, and hyperparameters $\lambda \in \mathbb{R}^+$, $L \in \mathbb{R}$ as

$$f_{\theta,\mathcal{B}}^{MC}(x) = \underbrace{f^{NN}(x)\left(e^{-\lambda\,d(x,s')}\right)}_{\text{Nearest Memory Neighbour Function}} + \underbrace{L\left(1 - e^{-\lambda\,d(x,s')}\right)\sigma(f^\theta(x))}_{\text{Constrained Neural Network Function Class}} \quad (2)$$

where, $s' = C_{\mathcal{B}|_S}(x)$ is the nearest memory to $x$, and $\sigma : \mathbb{R} \mapsto [-1,1]$ is a compressive non-linearity that imposes hard limits on the outputs of $f^\theta$. In practice, we use $\tanh$ or similar functions.

We refer the reader to Figure 2 to drive the intuition. For inputs that are close to the points in the codebook $\mathcal{B}$, the function predicts values that are similar to the one observed in the training dataset. More concretely, the value predicted by the function is a simple mixture: $\alpha f^{NN}(x) + (1-\alpha)\,L\,\sigma(f^\theta(x))$, where the mixing factor $\alpha \in [0,1]$ changes in proportion to the distance to the nearest memory. Thus, for points further away more weight is placed on the neural network and the memories have little influence. The degree of permissible deviation from nearest neighbor prediction $f^{NN}$ is controlled by the parameter $\lambda$. Thus, we obtain a purely nearest neighbor function for $\lambda = 0$, and a vanilla deep neural network function for $\lambda = \infty$. Note that the MCNN function values in regions far away from memories are in the set $[-L, L]$. For this reason, we normalize output actions to $[-L, L]$ before training an MCNN function.

### 4.2 THEORETICAL ANALYSIS OF MCNNS FOR IMITATION LEARNING

For fixed hyperparameters $L$, $\lambda$ and memory codebook $\mathcal{B}$, we denote by $\mathfrak{F}$, the class of memory-consistent functions outlined in Equation 2. Note that a choice of the DNN function parameters $\theta$ fixes a specific function in this class as well.

**Assumption 4.3** (Realizability). We assume that the expert policy $\pi^*$ belongs to the function class $\mathfrak{F}$.

This assumption trivially holds at the memories, where the MCNN exactly reproduces expert actions. For all other points, we assume that there exist some parameters $\theta$, which can capture the policy $\pi^*$ with sufficient accuracy, for a choice of $L$ and $\lambda$. For a point $x$ at a distance of $d(s', x)$, the vanilla DNN can affect the predictions only by an amount of $\pm L\left(1 - e^{-\lambda\,d(x,s')}\right)$. Without this restriction, we might have been able to capture behaviors that went well beyond these ranges. This is reasonable since, expert policies do not make sudden unbounded jumps in their actions. What we propose here is a way to enforce this bound using a zeroth order nearest neighbor estimate.

We analyze the behavior of this function class, and present some useful lemmas along the way. For a set of memories present in $\mathcal{B}|_S$, we wish to capture the maximum value that the distance term: $d(x, s')$ can take in Equation 2. To that end, we define the *most isolated state* as the following:

**Definition 4.4** (Most Isolated State). For a given set of memory points $\mathcal{B}|_S$, we define the most isolated state $s^I_{\mathcal{B}|_S} := \arg\max_{s \in S} \left( \min_{m \in \mathcal{B}|_S} d(s, m) \right)$, and consequently the distance of the most isolated point as $d^I_{\mathcal{B}|_S} = \min_{m \in \mathcal{B}|_S} d(s^I_{\mathcal{B}|_S}, m)$

The distance of the most isolated state captures the degree of emptiness that persists with the current knowledge of the state space due to memory code-book $\mathcal{B}$.

**Lemma 4.5.** *Assume two sets of memory code-books $\mathcal{B}_i$, $\mathcal{B}_j$, such that $\mathcal{B}_i \subseteq \mathcal{B}_j$, then $d^I_{\mathcal{B}_i|_S} \geq d^I_{\mathcal{B}_j|_S}$*

*Proof:* The proof of the above lemma is straightforward, since the infimum of a subset ($\mathcal{B}_i$) is larger than the infimum of the original set ($\mathcal{B}_j$). $\qquad\square$

This observation is useful when we study the effects of increasing the size of the code-book $\mathcal{B}$. Note, when learning a memory-consistent neural network $f^{MC}_{\theta,\mathcal{B}}$, we deploy the standard SGD based training to adjust the parameters $\theta$. The choice of the number of memories in $\mathcal{B}$ is kept as a hyperparameter. This allows us to bound the maximum width of the function class, first described in [25] . We analyze this for single output functions next.

**Lemma 4.6.** *(Width of Function Class) The width of the function class $\mathfrak{F}$, $\forall \theta_1, \theta_2 \in \Theta$, and $\forall s \in \mathcal{S}$, defined as $\max_{\theta_i, \theta_j} |\left( f^{MC}_{\theta_i,\mathcal{B}} - f^{MC}_{\theta_j,\mathcal{B}} \right)(s)|$ is upper bounded by : $2L \times \left( 1 - e^{-\lambda\, d^I_{\mathcal{B}|_S}} \right)$*

*Proof*: Please see Appendix B.

**Theorem 4.7.** *The sub-optimality gap $J(\pi^*) - J(\hat{\pi}) \leq min\{H, H^2|\mathcal{A}|L\left(1 - e^{-\lambda\, d^I_{\mathcal{B}|_S}}\right)\}$*

*Proof:* Please see Appendix C.

**Corollary 4.8.** *Using Lemma 4.5 we know that if $\mathcal{B}_i \subseteq \mathcal{B}_j$, then $\left(1 - e^{-\lambda\, d^I_{\mathcal{B}_i|_S}}\right) \geq \left(1 - e^{-\lambda\, d^I_{\mathcal{B}_j|_S}}\right)$. This can result in lower performance gap according to Theorem 4.7, when $H \geq H^2|\mathcal{A}|L\left(1 - e^{-\lambda\, d^I_{\mathcal{B}|_S}}\right)$. Hence, reflecting the utility of adding more memories in such cases.*

**Takeaways.** We summarize the insights from the above theoretical analysis here. First, *our MCNN class of functions is bounded in* width *(Lemma 4.6)* even though it uses a high-capacity function approximator like DNNs. *No such bound is available for vanilla neural networks*. This translates to a *bounded sub-optimality gap (Theorem 4.7) also not available in vanilla neural networks*. Finally, our Corollary states that we can likely gain better imitation learning performance by simply adding more memories (up to a limit).

### 4.3 ALGORITHM: IMITATION LEARNING WITH MCNN POLICIES

We now describe our algorithm to use MCNNs for imitation learning. The first step in our method is to learn the memory code-book $\mathcal{B}$ from the expert trajectory dataset $D$. The goal of Algorithm 1 is to build the nearest memory neighbor function $f^{NN}$. This is followed by details on the training aspects of the MCNN parameters from the imitation dataset $\mathcal{D}_e$ in Algorithm 2.

For building the memory code-book, we leverage an off-the-shelf approach, Neural Gas [8], that selects prototype samples to summarize a dataset. For completeness, we summarize this approach briefly below.

**Definition 4.9** (Neural Gas). A neural gas $\mathcal{G} := (\mathcal{N}, \mathcal{E})$, is composed of the following components,

1. A set $\mathcal{N} \subset \mathcal{S}$ of the nodes of a network. Each node $m_i \in \mathcal{N}$ is called a memory in this paper.
2. A set $\mathcal{E} \subset \{(m_i, m_j) \in \mathcal{N}^2, i \neq j\}$ of edges among pairs of nodes, which encode the topological structure of the data. The edges are unweighted.

**Neural gas.** The neural gas algorithm [8, 30, 22] is primarily used for unsupervised learning tasks, particularly for data compression or vector quantization. The goal is to group similar data points together based on their similarities. The algorithm works by creating a set of prototype vectors, also known as codebook vectors or neurons. These vectors represent the clusters in the data space. The algorithm works by adaptively placing prototype vectors in the data space and distributing them like

---

**Algorithm 1** Learning Memories

**Input:** Offline dataset $\mathcal{D} = \{(s_i, a_i)\}_{i=1}^N$ , number of memories $m$
**Output:** A nearest neighbor based function $f^{NN} : \mathcal{S} \mapsto \mathcal{A}$

1: Nodes $\mathcal{N}$, edges $\mathcal{E} \leftarrow$ NeuralGasClustering$(\mathcal{S}, m)$          // learns the distribution induced by $\mathcal{D}$
2: Nodes $\mathcal{N}'$, $\mathcal{D}(\mathcal{N}') \leftarrow$ For each node in $\mathcal{N}$, find the closest observation in $\mathcal{D}$, and call this $\mathcal{N}'$. Additionally, return the corresponding action taken by the expert in $\mathcal{D}$, denoted by the map $\mathcal{D}(\mathcal{N}')$
3: $\mathcal{G} \leftarrow$ Define neural-gas with nodes $\mathcal{N}'$, and edges $\mathcal{E}$.
4: Define a memory code-book $\mathcal{B}$ using $\mathcal{B}|_{\mathcal{S}} = \mathcal{G}$ and $\mathcal{B}|_{\mathcal{A}} = \mathcal{D}(\mathcal{N}')$. Pairing nodes in the neural-gas to its corresponding actions.
5: Define a nearest neighbor function $f_{\mathcal{B}}^{NN}$, along the lines described in Definition 4.1 using $\mathcal{B}$.
6: **return** $f_{\mathcal{B}}^{NN}$

---

a *gas* in order to capture the density. For more details we refer the reader to [8]. We use this in Algorithm 1 (Line 1) to get the initial clustering. We can now go ahead and outline how "memories" are picked in our case.

**Learning memories.** Algorithm 1 first uses the neural-gas algorithm to pick candidate points (nodes) $\mathcal{N}$ in the state space. However, these points could be potentially absent in the dataset $\mathcal{D}$, making it hard to associate the correct action. To remedy this situation, we replace these points with the closest states from the training set as *memories*. Such *memory* states come with the corresponding actions taken by the expert. This is then used to define a nearest neighbor function by building the memory codebook $\mathcal{B}$ and defining a function as outlined in Definition 4.1.

---

**Algorithm 2** Behavior Cloning with Memory-Consistent Neural Networks: Training

**Input:** Dataset $\mathcal{D} = \{(s_i, a_i)\}_{i=1}^N$, nearest neighbor function $f_{\mathcal{B}}^{NN}$, neural network function $f^\theta(.)$, batch size, total training steps $T$, parameters $\lambda$ and $L$.
**Output:** Learned policy $f_{\theta,\mathcal{B}}^{MC}$

1: **for** step= 1 to $T$ **do**
2:      Sample batch $B$ from $\mathcal{D}$.
3:      Forward propagate $(s_i, a_i) \sim B$, $f_{\theta,\mathcal{B}}^{MC}(x) = f_{\mathcal{B}}^{NN}(x) \left( e^{-\lambda\, d(x,s')} \right) + L \left( 1 - e^{-\lambda\, d(x,s')} \right)\, \sigma(f^\theta(x))$

     where, $s'$ is the nearest neighbor of $x$ in $\mathcal{B}|_S$, $\sigma_\beta(x)$ is a tanh-like activation function given by $\sigma_\beta(x) = 2 \left[ \text{LeakyReLU}_\beta \left( \frac{x+1}{2} \right) - (1 - \beta)\text{ReLU}\left( \frac{x-1}{2} \right) \right] - 1$ and $\beta = \max\left( 0, 1 - \lfloor \frac{step}{100} \rfloor \right)$.
4:      Update $\theta \leftarrow \theta - \nabla\, \mathbb{E}_{(s_i, a_i) \sim B}\, \mathcal{L}(f_{\theta,\mathcal{B}}^{MC}(s_i), a_i)$ where $\mathcal{L}$ is the negative log-likelihood or mean squared loss or other loss function.
5: **end for**

---

**Training MCNNs.** Finally, we train MCNN policies through gradient descent on the parameters $\theta$ of the neural network over the expert dataset $D_e$. For the compressive non-linearity, we use the $\sigma_\beta$ function given in Algorithm 2 which is similar to $\tanh$. We describe this in detail in Appendix F.

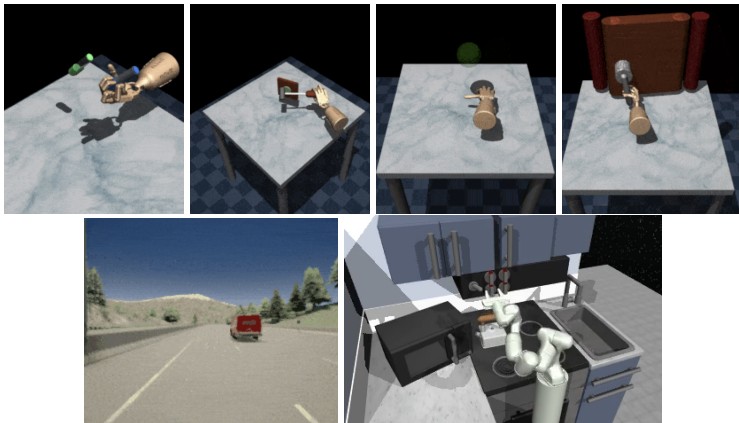

Figure 3: The environments here include: Adroit Pen, Hammer, Relocate, and Door [33], CARLA's Town03 and Town04 [3], and Franka Kitchen [11]. The four Adroit environments and Franka Kitchen have proprioceptive observations and the CARLA environment has image observations.

## 5 EXPERIMENTAL EVALUATION

We now perform a thorough experimental evaluation of MCNN-based behavior cloning in a large variety of imitation learning settings.

**Environments and Datasets:** We test our approach on 10 tasks, in 6 environments: 4 Adroit dextrous manipulation, 1 CARLA environment, and the Franka Kitchen environment as pictured in Figure 3. Demonstration datasets are drawn from D4RL [9] for Adroit and CARLA and from the multimodal relay policy learning dataset [11] for Franka Kitchen. For each Adroit task, we evaluate imitation learning from 2 different experts: (1) small realistic human demonstration datasets (**'human'**) with 25 trajectories per task (5000 transitions), and (2) large demonstration datasets with 5000 trajectories (1 million transitions) from a well-trained RL policy (**'expert'**). In CARLA, we train on demonstrations from a hand-coded expert. The Franka Kitchen demos were collected by humans wearing VR headsets. For observations, we use high-dimensional states in Adroit and Franka Kitchen, and $48 \times 48$ images in CARLA. Action spaces are 24-30 dimensional in the Adroit, 9-D in Franka Kitchen, and 2-D in CARLA. Further, in all Adroit environments, a goal is randomly chosen at reset and goal information is included in the observation vector (more in Appendix F).

**Baselines:** We run the following baselines for comparison. **(1) Behavior Cloning:** We obtain results with a vanilla MLP architecture. The details of the architecture can be found in Appendix F. We report results from our implementation of BC and also report results given in D4RL [9] under the names '**MLP-BC**' and '**D4RL BC**' respectively. Our BC implementation has only one difference from [9]'s implementation: we normalize the observations. Normalizing observations has been shown to improve BC's performance [10]. **(2) 1 Nearest Neighbours (1-NN):** We set up a simple baseline where the action for any observation in the online evaluation is the action of the closest observation in the training data. In the expert and cloned datasets for each environment, this amounts to having to perform a search amongst a million datapoints online at every step (which is highly inefficient). **(3) Visual Imitation with (k) Nearest Neighbours (VINN)** [26]: VINN is a recent method that performs a Euclidean kernel weighted average of some k nearest neighbors. In the Adroit case, we directly perform the k nearest neighbors on the raw observation vectors. In the CARLA case, we perform it in the same embedding space that we use to create memories (we discuss this embedding space more below). **(4) CQL-Sparse (CQL-S):** We learn a policy using the CQL offline RL algorithm [18] and a sparse reward given for task completion only. **(5) Implicit BC (IBC)** [7]: We report the results from [7] which performs BC with energy models on the human tasks. **(6) Behavior Transformer (BeT-BC)** [38]: We train and evaluate a behavior transformer using the official implementation on all tasks. **(7) Diffusion BC (Diff-BC)** [44, 1]: We also train and evaluate a diffusion-based BC policy using the implementations in [44, 1]. We provide additional details for all baselines and comprehensive hyperparameter sweeps in Appendix F.

**Learning memories and MCNN:** We learn memories using the incremental neural gas algorithm for 10 epochs starting from $2.5\%$ of the total dataset to $10\%$ of the total dataset for each task. We update all the transitions in each dataset by appending the closest memory observation and its corresponding target action ( Algorithm 1). We train the MCNN on this dataset following Algorithm 2 for 1 million steps and evaluate on 20 trajectories after training and repeat each experiment for a minimum of 3 seeds. We report results with an MLP, a behavior transformer (BeT), and a diffusion policy as the underlying neural network under the names **'MCNN+MLP'**, **'MCNN+BeT'**, and **'MCNN+Diff'** respectively. We expand on the experimental setup and all hyperparameters in Appendix F.

**Embedding CARLA images:** In the CARLA tasks, we use an off-the-shelf ResNet34 encoder [2] that has been shown to be robust to background and environment changes in CARLA to convert the $48 \times 48$ images to embeddings of size 512. We use this embedding space as the observation space for learning memories and policies.

**Performance Metrics:** All our environments come with pre-specified dense task rewards which we use to define performance metrics. We report the cumulative rewards (return) for each task. For the aggregate plots on a set of four tasks, we compute the percentage increase in return of a method over D4RL BC in all four tasks and report the median.

**Results:** First, for the "human" tasks with the most realistic imitation learning setup, we plot aggregate and taskwise results in Figure 4. In aggregate-human, we see that MCNN+MLP with fixed hyperparameters performs the best followed by MCNN+Diff and IBC at second place. We also see that the MCNN variants of MLP, BeT and Diffusion consistently outperform the vanilla versions.

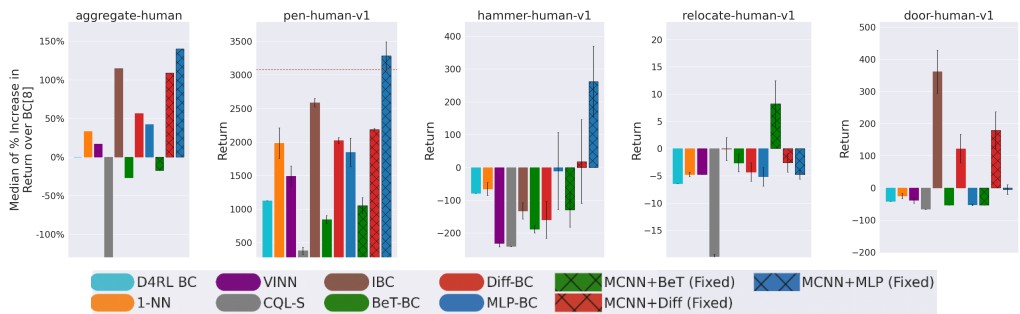

Figure 4: **Adroit human tasks [25 demos]:** Comparison of returns (across 20 evaluation trajectories and 3 random seeds) between baselines and our methods (MCNN+BeT, MCNN+Diff, and MCNN+MLP). Our MCNN methods use the same fixed set of hyperparameters across all tasks.

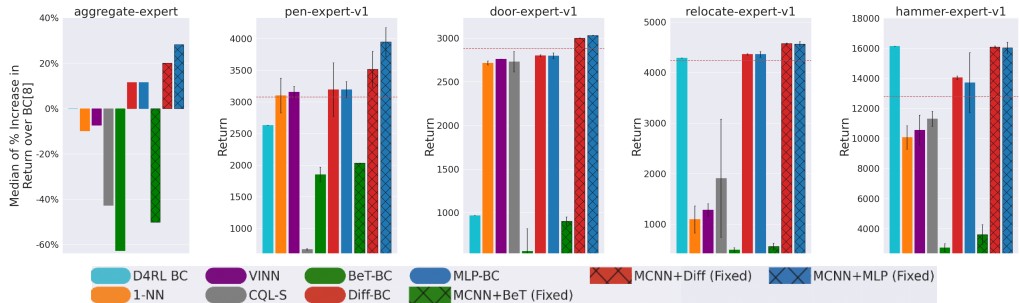

Figure 5: **Adroit expert tasks [5000 demos]:** Comparison of returns (across 20 evaluation trajectories and 3 random seeds) between baselines and our methods (MCNN+BeT, MCNN+Diff, and MCNN+MLP). Our MCNN methods use the same fixed set of hyperparameters across all tasks.

In pen-human-v1, MCNN+MLP outperforms the nearest baseline by 33%. It is also the only method to shoot past the expert ceiling of 100 (depicted by a dashed red line). In hammer-human-v1, MCNN+MLP is the only method to obtain a positive return outpacing the nearest baseline by an order of magnitude (from -11 to 262). In relocate-human-v1, MCNN+BeT is the only method to achieve a positive return. We attribute, like previous work [24], the stronger performance of MCNN+BeT over other MCNN variants in the relocate task to the 'memory' advantage available to transformers that is specifically suited for this task (where the historical states inform whether the ball has been grasped). Lastly, in door-human-v1, MCNN+Diff outperforms all but the IBC baseline by an order of magnitude. In this task, where repeated attempts at grasping and opening the door handle are usually required for success, we see methods that enable such repetition (energy models in IBC and diffusion in MCNN+Diff) succeed.

On the expert tasks in Figure 5, even with a large amount of data, we see MCNN+MLP come in first outperforming the nearest baseline in the aggregate plot by over 100%. MCNN+Diff comes in second in aggregate with a 40% improvement over the nearest baseline. Here too, MCNN variants outperform the vanilla versions across all tasks. Also, MCNN+MLP and MCNN+Diff are the only methods to exceed the expert ceiling in all four tasks. Across expert tasks both MCNN+MLP and MCNN+Diff perform competitively and obtain up to a 25% improvement over the nearest baseline.

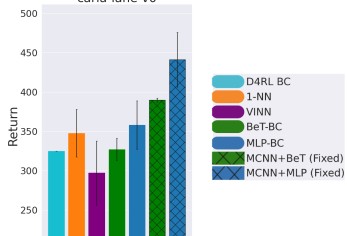

Figure 8: **CARLA [400 demos]:** Comparison of return (across 20 evaluation trajectories and 3 random seeds) between baselines and our methods (MCNN+BeT and MCNN+MLP). For the results shown here, our MCNN methods use the same fixed set of hyperparameters across all tasks.

Table 1: **Franka Kitchen [566 demos]:** Comparison of probabilities/success rates of interacting with 1 to 5 objects (given by p1 to p5 respectively) in Franka Kitchen between baselines and one of our methods (MCNN+Diff).

|  | Franka Kitchen | | | | |
|---|---|---|---|---|---|
|  | p1 | p2 | p3 | p4 | p5 |
| LSTM-GMM [21] | **1.0** | 0.9 | 0.74 | 0.34 | 0 |
| IBC | 0.99 | 0.87 | 0.61 | 0.24 | 0 |
| BeT-BC | 0.99 | 0.93 | 0.71 | 0.44 | 0 |
| Diff-BC [1] | **1.0** | **1.0** | **1.0** | 0.99 | 0.02 |
| MCNN+Diff | **1.0** | **1.0** | **1.0** | **1.0** | **0.08** |

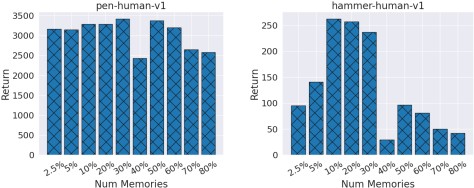

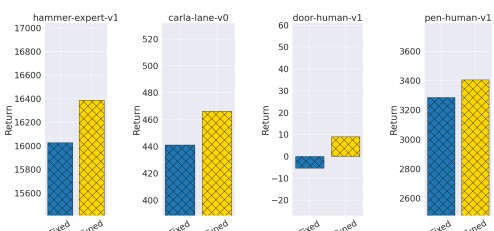

Figure 6: Number of Memories in MCNN+MLP: Comparison of returns (across 20 evaluation trajectories) with our method (MCNN+MLP) using between 2.5% to 80% of the dataset as memories. We find a "sweet spot" for number of memories at 10-20%. We also see the expected decrease to 1-NN performance as number of memories increases to 100%.

Figure 7: Fixed vs Tuned Hyperparameters in MCNN+MLP: Comparison of returns (across 20 eval trajectories and 3 seeds) between our method with fixed hyperparameters across all tasks and with hyperparameters tuned online. Hence, it's possible to improve performance with limited online interaction.

In the high dimensional CARLA lane task as well, we see in Figure 8 that MCNN+MLP outperforms the nearest baseline by 27%. MCNN+BeT comes in a close second.

In Table 1, we compare MCNN+Diff with various baselines in the multimodal Franka Kitchen task. Following previous work [1], we compare the probability of success in interacting with 1 to 5 objects (p1-p5). MCNN+Diff performs similarly to Diff-BC [1] and obtains 100% success on p1 to p4. On p5, where baselines have failed to show any success, MCNN+Diff obtains a 4x improvement over the nearest baseline. Additional figures and all means and standard deviations of our results can be found in Appendix G.

**Discussion:** We identify some high-level trends across our results here. For every architecture – MLP, BeT, or Diffusion, plugging in MCNN significantly improves performance in every task. The best-performing method in nearly every task is an MCNN-based method. MCNN can even improve the performance of simple MLP architectures to beyond that of more sophisticated recent architectures such as Diffusion models. For example, in pen-human-v1 in Figure 4, diffusion outperforms MLP but MCNN+MLP significantly improves upon Diffusion. The above statements remain true across different types and sizes of expert data and across disparate tasks. We discuss some ablations next.

**Ablations:** We plot return against number of memories in Figure 6 for MCNN+MLP. It demonstrates the existence of a "sweet spot" for the number of memories around 10-20% of the dataset. This allows for more efficient inference in MCNN than in baselines like VINN and 1-NN. It also shows the expected degradation to 1-NN performance as the number of memories increases towards 100%. Additional discussion on computation cost and improved efficiency of MCNN compared to VINN and 1-NN can be found in Appendix F.

In Figure 7, we show that given limited online interaction (20 episodes), selecting the best-performing hyperparameters ($\lambda$, $L$, and number of memories) further improves MCNN+MLP performance. We show the significance of neural gas-based memories by comparing MCNN+MLP with a version that uses randomly chosen memories in Figure 13 in Appendix G. We observe significant reduction in performance with randomly chosen memories. Neural-gas uses competitive Hebbian learning algorithm which is better at capturing the distribution of training points (creating memories that are "spread out"). This reduces the distance to the most isolated state, improving imitation performance. Ablation results on the values of $\lambda$ and other tasks is in Appendix G.

## 6 Conclusions and Limitations.

Imitation learning, and in particular behavior cloning, is one of the most promising approaches when it comes to transferring complex robotic manipulation skills from experts to embodied agents. In this work, we introduced MCNNs, a semi-parametric approach to behavior cloning that significantly increases the performance of behavior cloning methods across diverse realistic tasks and datasets regardless of the underlying architecture (MLP, transformer, or diffusion). While our theoretical and empirical results support the idea that appropriately constraining the function class based on training data memories improves imitation performance, MCNNs are only one heuristic way to accomplish this; it is very likely that there are even better-designed model classes in this spirit, that we have not explored in this work. Finally, we would like in future work to explore MCNNs as model classes beyond just behavior cloning, such as in reinforcement learning and meta-learning.

**Acknowledgements:** This work was supported in part by ARO MURI W911NF-20-1-0080, NSF 2143274, and ONR N00014-22-1-2677. Any opinions, findings, conclusions or recommendations expressed in this material are those of the authors and do not necessarily reflect the views the Army Research Office (ARO), Office of Naval Research (ONR), the Department of Defense, or the United States Government.

**Reproducibility Statement:** To ensure reproducibility, we have released all code for MCNN variants and the baselines at our website https://sites.google.com/view/mcnn-imitation. We have also described all environments, datasets, baselines, and MCNN variants in detail (with their hyperparameters) in Section 5 and Appendix F.

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

APPENDIX

## A EXAMPLES OF THE MCNN FUNCTION CLASSES

We demonstrate the effects of varying number of memories, $\lambda$, and L in Figures 9, 10, and 11 respectively. By increasing L or decreasing the number of memories, we directly increase the width of the model class. By increasing $\lambda$, we quicken the interpolation from the nearest neighbor components to the neural network function class. Similarly, by decreasing $\lambda$, we slow this transition.

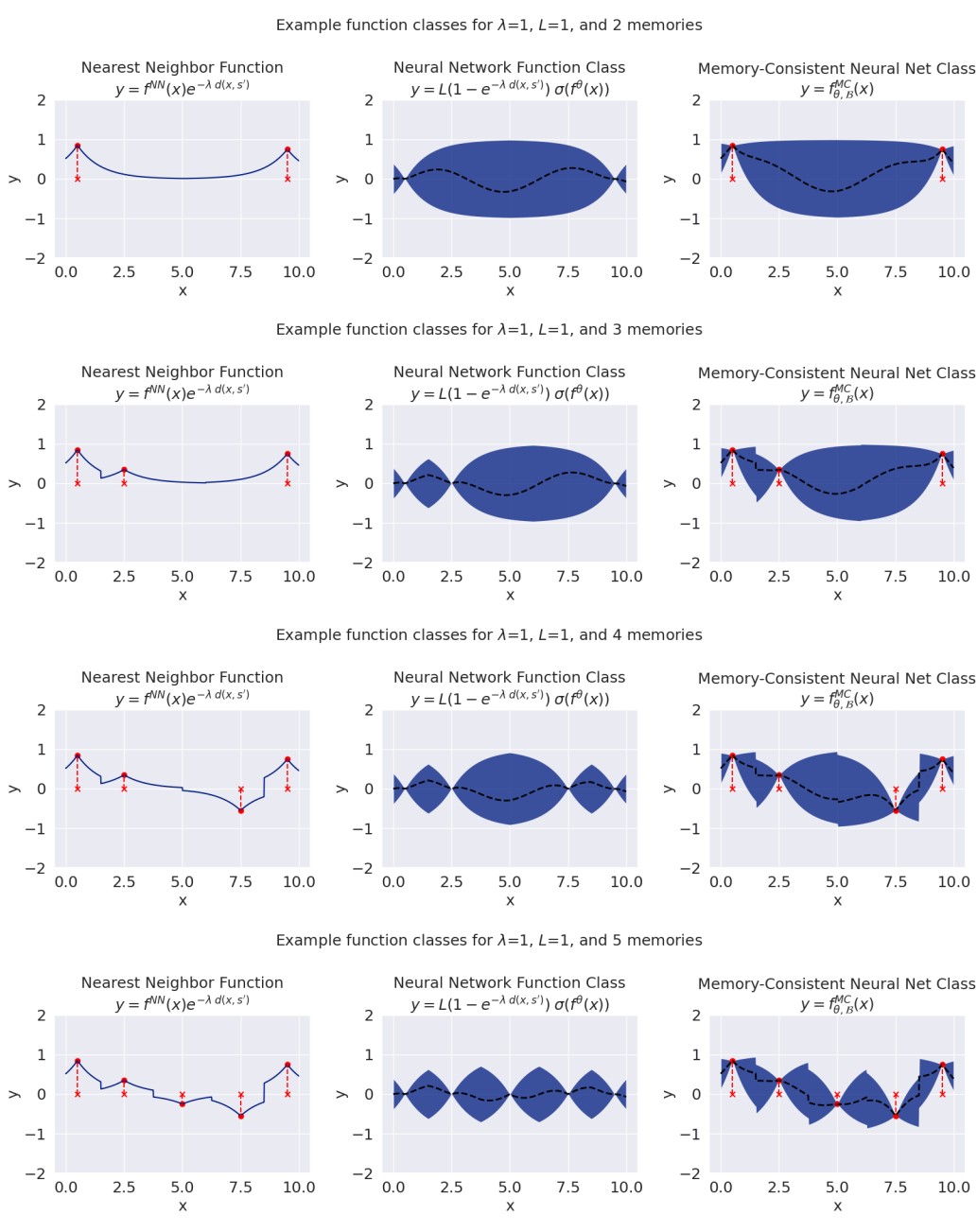

Figure 9: Effects of varying number of memories (keeping $\lambda$ and L fixed) on the MCNN function class.

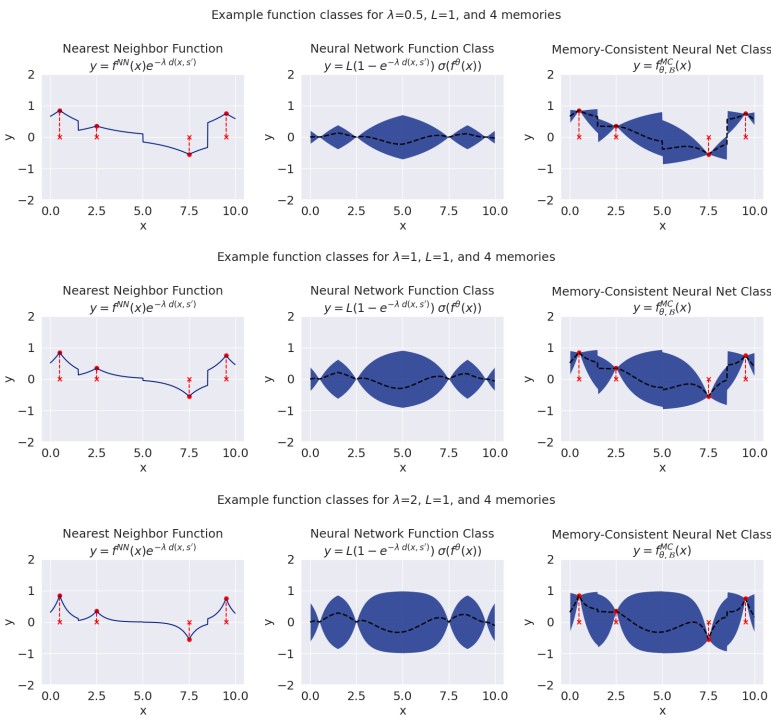

Figure 10: Effects of varying $\lambda$ (keeping L and number of memories fixed) on the MCNN function class.

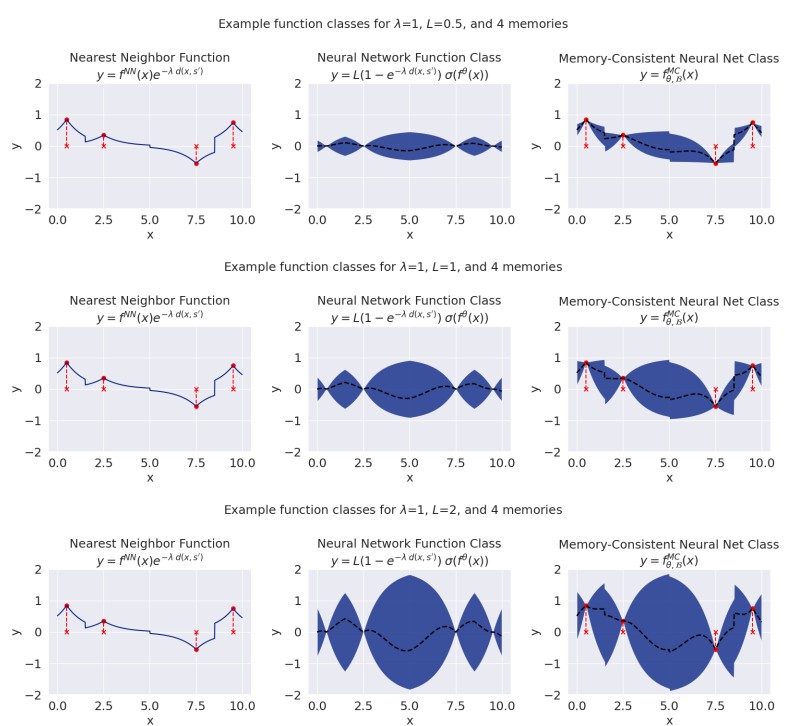

Figure 11: Effects of varying L (keeping $\lambda$ and number of memories fixed) on the MCNN function class.

## B    PROOF OF LEMMA 4.6

**The width of the function class $\mathfrak{F}$, $\forall \theta_1, \theta_2 \in \Theta$, and $\forall s \in \mathcal{S}$, defined as $\max\limits_{\theta_i, \theta_j} |\left(\mathbf{f}^{MC}_{\theta_i, \mathcal{B}} - \mathbf{f}^{MC}_{\theta_j, \mathcal{B}}\right)(s)|$ is upper bounded by $2L \times \left(1 - e^{-\lambda\, d^I_{\mathcal{B}|s}}\right)$**

*Proof.* Using Equation 2, we can express the difference as the following :

$$|f^{MC}_{\theta_i, \mathcal{B}} - f^{MC}_{\theta_j, \mathcal{B}}|(x) = |L\left(1 - e^{-\lambda\, d(x, s')}\right)\, \sigma(f^{\theta_i}(x)) - L\left(1 - e^{-\lambda\, d(x, s')}\right)\, \sigma(f^{\theta_j}(x))| \quad (3)$$

$$= L\left(1 - e^{-\lambda\, d(x, s')}\right)|\sigma(f^{\theta_i}(x)) - \sigma(f^{\theta_j}(x))| \quad (4)$$

$$\leq L\left(1 - e^{-\lambda\, d^I_{\mathcal{B}|S}}\right)|\sigma(f^{\theta_i}(x)) - \sigma(f^{\theta_j}(x))| \quad (5)$$

Now, observing that $|\sigma(f^{\theta_i}(x)) - \sigma(f^{\theta_j}(x))| \leq 2$ completes the proof. $\qquad\square$

## C    PROOF OF THEOREM 4.7

**The sub-optimality gap $\mathbf{J}(\pi^*) - \mathbf{J}(\hat{\pi}) \leq \min\{\mathbf{H}, \mathbf{H^2}|\mathcal{A}|\mathbf{L}\left(1 - \mathbf{e}^{-\lambda\, \mathbf{d^I_{\mathcal{B}|s}}}\right)\}$**

*Proof.* In order to take advantage of well-known results in the imitation learning literature [36, 31, 2, 32], we restrict ourselves for the purpose of this analysis to the discrete action-space $\mathcal{A}$ scenario, where the policy $\pi : \mathcal{S} \mapsto \Delta(\mathcal{A})$. Even still, the intuitions developed through this analysis guide our algorithmic choices in continuous environments. The actions picked in the expert dataset $D$ induce a dirac distribution over the actions corresponding to each input state.

Recall that in imitation learning, if the population total variation (TV) risk $\mathbb{T}(\hat{\pi}, \pi^*) \leq \epsilon$, then, $J(\pi^*) - J(\hat{\pi}) \leq min\{H, H^2\epsilon\}$ (See [31] Lemma 4.3). We note the following for population TV risk:

$$\mathbb{T}(\hat{\pi}, \pi^*) = \frac{1}{H}\sum_{t=1}^{H}\mathbb{E}_{s_t \sim f^t_{\pi^*}}\left[TV(\hat{\pi}(\cdot|s_t), \pi^*(\cdot|s_t))\right] \leq \frac{1}{H}\sum_{t=1}^{H}\mathbb{E}_{s_t \sim f^t_{\pi^*}}\left[|\mathcal{A}|L\left(1 - e^{-\lambda\, d^I_{\mathcal{B}|S}}\right)\right]$$

$$\leq |\mathcal{A}|L\left(1 - e^{-\lambda\, d^I_{\mathcal{B}|S}}\right)$$

$$(6)$$

where $f^t_{\pi^*}$ is the empirical distribution induced on state $s^t$ obtained by rolling out policy $\pi^*$. For the first inequality in the above derivation, we use Lemma 4.6. Using this, in the performance gap lemma gives us the following:

$$J(\pi^*) - J(\hat{\pi}) \leq min\{H, H^2|\mathcal{A}|L\left(1 - e^{-\lambda\, d^I_{\mathcal{B}|S}}\right)\}$$

$$\square$$

## D    EXTENDED RELATED WORK

**Compounding errors in imitation learning** have previously been tackled by permitting online experience [13, 34], reward labels [28], queryable experts [37], or modifying the demonstration data collection procedure [19]. Our work is orthogonal to these methods and creates a model class that avoids compounding errors by construction. Other works that propose new models for IL such as Implicit BC (IBC) [7], Behavior Transformer (BeT) [38], Action Chunking Transformer [48], and Diffusion Policies [44, 1, 27] are orthogonal to our approach. MCNN can be used as a plug-in approach to improve any of these methods. In fact, we show that MCNN with a BeT backbone outperforms vanilla BeT and MCNN with a diffusion model outperforms diffusion BC on all tasks in our experiments in Section 5. We also show that MCNN outperforms IBC in Section 5.

**Non-parametric and semi-parametric methods in imitation learning** such as nearest neighbors [39], RBFs [35], and SVMs [20] have historically shown competitive performance on various robotic control benchmarks. But, only recently, a semi-parametric approach consisting of neural networks for representation learning and k-nearest neighbors for control was proposed in Visual Imitation through

Nearest Neighbors (VINN) [26]. This is the closest paper to our work and in Section 5, we compare with VINN and demonstrate that we outperform their method comprehensively.

**Theoretical guarantees on the sub-optimality gap in imitation learning** with MCNN are provided in this paper. Such guarantees are not available with vanilla neural networks. Our theorem builds on earlier work on reductions for imitation learning in [36, 31, 2, 32] and leverages intuitions from [25] on bounding the width of the model class.

**Learning a codebook of prototypes**, like our memories, has been previously explored for image reconstruction [43, 6], physics-constrained learning [40], online RL [23, 47, 12, 5], interpretable OOD detection [45, 16, 17, 15], robust classification [4, 14, 41], and motion prediction [46]. The closest related usage of memories that are representative of the topology of the input space is in [40]. But, here, external information in the form of physics and medical constraints plays a key role in enforcing constraints at these pivotal points. In this paper, our prior is simply a kind of "consistency" with no external information utilized.

# E COMPUTE

We ran the experiments on either two Nvidia GeForce RTX 3090 GPUs (each with 24 GB of memory) or two Nvidia Quadro RTX 6000 GPUs (each with 24 GB of memory). The CPUs used were Intel Xeon Gold processors @ 3 GHz. Videos of our policies for many random goals across environments as well as our code can be found in our website https://sites.google.com/view/mcnn-imitation.

# F DETAILED EXPERIMENTAL SETUP

**Additional information about our task definition:** Following notation in prior works [9, 18, 7], we define an imitation learning task as a tuple of (environment, reward function, and demonstration dataset). Since we perform experiments with both "expert" and "human" demonstrations, our experiment list has the following 9 tasks: pen-human-v1, pen-expert-v1, hammer-human-v1, hammer-expert-v1, door-human-v1, door-expert-v1, relocate-human-v1, relocate-expert-v1, and carla-lane-v0.

**Additional information for Adroit environments:** All policies learned in the four Adroit environments are goal-conditioned. The goal positions and orientations are provided as part of the observation vector. Every time that the environment is reset (such as for a new evaluation rollout), the goal is randomly chosen. These include random goal orientations of the pen, goal locations for the relocate task, door locations in the door task, and nut-and-bolt location in the hammer task. In Adroit human tasks in particular, with only 25 demonstrations, it is not possible to cover every random goal location for each of the four environments (such as every possible pen orientation) in the training dataset. This increases the difficulty of the four adroit tasks. Even with this challenge, MCNN methods outperform baselines across environments and demonstrate their ability to generalize even from small datasets of demonstrations.

We also provide videos of the Adroit human demos in our website. The multimodality in these datasets collected by humans wearing VR gloves is visible in these videos. Some examples include (1) in the hammer task, some demonstrations move the hammer above the bolt before hitting it, some move it below, and some directly hit the bolt and (2) in the pen task, some demonstrations have the little finger above the pen and some have it below the pen throughout the episode. MCNN helps address the significant compounding error challenge which MLP-BC fails to address and hence results in MCNN+MLP having an overall better performance than Diffusion-BC even though Diffusion is better equipped to handle multimodality [1]. Our results in Franka Kitchen (see Appendix G) highlight this where MCNN+Diffusion improves upon the already strong Diffusion-BC baseline.

**Additional information for Franka Kitchen environment:** In this environment, a Franka robot arm interacts with various items in a Kitchen such as a kettle, microwave, light switch, cabinets, burners, etc. We use the multimodal relay policy learning dataset [11] with 566 demonstrations collected by humans wearing VR headsets interacting with 4 objects (out of a total 7 in the environment) in each episode. The goal is to execute as many demonstrated tasks as possible and the metrics capture the probability of successful interactions with various numbers of objects. These metrics, following

previous work [1], have been named p1 to p5 for the probability of success in interacting with 1 to 5 objects respectively.

**Normalization of inputs:** We normalized all observations by subtracting the mean and dividing by standard deviation. We didn't have to normalize the actions as they are already in the range $[-1, 1]$ but if we are learning transition models instead, the outputs (next state or reward) would have to be normalized.

**Our** $\tanh$**-like dynamic activation function:** We plot our activation function $\sigma_\beta(.)$ described in Algorithm 2 in Figure 12. It is exactly like the $\tanh$ function for $\beta = 0$ as seen in Figure 2. Further, as given in Algorithm 2's Line 3, we set $\beta = \max\left(0, 1 - \lfloor\frac{step}{100}\rfloor\right)$. Hence, $\beta = 0$ after 100 steps until 1 million steps during training. We also set $\beta = 0$ during inference. For $\beta = 0$, our activation function reduces to $-1$ for $x < -1$, $x$ for $-1 \le x \le 1$, and 1 for $x > 1$. This is exactly like $\tanh$. The reason for using this activation function is the very few initial steps when it is not like $\tanh$ where gradients are available beyond $[-1, 1]$ during which time, the neural network component adjusts for the presence of the nearest neighbor component. We use the standard $tanh$ activation function for our reimplementation of BC.

**Implementation details for baselines and MCNN+MLP:** In BC and MCNN+MLP, we use an MLP with two hidden layers (three total layers) of size $[256, 256]$ for Adroit tasks and $[1024, 1024]$ for CARLA. We use an Adam optimizer with a starting learning rate of $3e - 4$ and train for 1 million steps. We simply minimize the mean squared error for training the policies. We use a batch size of 256 throughout. We describe the MCNN-specific hyperparameters, namely $\lambda$, $L$, and number of memories in another paragraph below. For all other BC hyperparameters, we use the recommended values in the TD3-BC implementation in [42].

For VINN, we set $k = 10$ in the Euclidean weighted k nearest neighbors algorithm. This value was recommended by the original paper [26].

For BeT, we follow the official implementation[2]. We use 6 layers, 6 heads, and an embedding dimension of 120 in the transformer model. We also 64 clusters for action discretization performed on the actions seen in the first 100 steps. Since the original paper [38] did not run experiments on D4RL tasks, we ran a sweep over various choices of number of layers (4, 6), number of heads (4, 6), and embedding dimension (32, 64, 120). Following previous work [7], we chose the hyperparameters (6 layers, 6 heads, 120 embedding dimension) with the highest average scores on three human tasks. We used these hyperparameters on all other tasks as well. For all other BeT hyperparameters, we use the recommended values in the official BeT implementation.

For Diffusion-BC, we use the official implementation from [44][3] in Adroit tasks and the official implementation from [1][4] in Franka Kitchen tasks. We also use the recommended hyperparameters provided in the code for pen-human-v1 in the former repository in all human and expert Adroit tasks. We similarly use the recommended hyperparameters provided in the latter repository along for the kitchen task.

In CQL-Sparse, we use an MLP with three hidden layers (three total layers) of size $[256, 256, 256]$ for Adroit tasks. We give a reward of 1 for the last timestep in both expert and human datasets where each trajectory achieves task completion. Hence, we run CQL-Sparse only on the human and expert datasets where each trajectory has achieved task completion. We use a reward of 0 in all preceding timesteps. For all CQL-sparse hyperparameters, we use the recommended values in [18]. We also ran a sweep over the three key hyperparameters of CQL, namely actor learning rate ($3e - 5, 1e - 4$), initial $\alpha$ (5, 10), and Lagrange threshold (5, None). We note that while performing the sweep, if the Lagrange threshold is None, CQL was run with fixed $\alpha$. Otherwise, $\alpha$ is tuned automatically, as described in Kumar et al. [18], based on the Lagrange threshold and starting from its initial value. We found that the recommended values in [18], *i.e.* actor learning rate = $3e - 5$, initial $\alpha = 5$, and Lagrange threshold = 5 performed the best overall.

**Implementation details for MCNN + Behavior Transformer (BeT):** We train MCNN+BeT following the official BeT implementation described above with one major change to the action offsets output by the BeT model. Let us denote the sequence of input observations as $\tau$ and the action

---

[2] https://github.com/notmahi/miniBET
[3] https://github.com/Zhendong-Wang/Diffusion-Policies-for-Offline-RL
[4] https://github.com/real-stanford/diffusion_policy

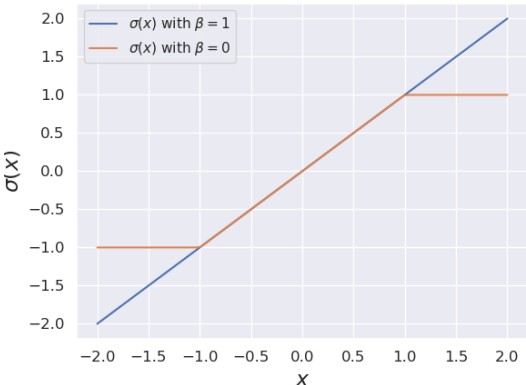

Figure 12: Dynamic $\mathrm{tanh}$-like activation function $\sigma_\beta(x)$ shown for $\beta = 0$ and $\beta = 1$.

offsets output by the BeT model as $f^{\mathrm{BeT}}(\tau)$. Then, the output of MCNN+BeT is given as follows:

$$f^{MC}_{\mathrm{BeT},\mathcal{B}}(\tau) = f^{NN}_{\mathcal{B}}(\tau) \left( e^{-\lambda \, d(\tau,\tau')} \right) + L \left( 1 - e^{-\lambda \, d(\tau,\tau')} \right) \, 2\sigma(f^{\mathrm{BeT}}(\tau)/2) \tag{7}$$

where $\tau'$ is the nearest (memory) sequence to $\tau$ and $f^{NN}_{\mathcal{B}}(\tau)$ retrieves the corresponding sequence of actions from the codebook. We also multiply and divide by two inside the $\mathrm{tanh}$-like function since each item in the sequence output by the BeT lies in $[-2, 2]$.

**Implementation details for MCNN + Diffusion:** We augment the diffusion process to use memories in every step. Rather than predicting the noise value, we predict the true values. This amounts to a minor change to the loss function within the official implementation. We provide this code in our repository.

**MCNN-specific hyperparameters:** We use a value of $L = 1.0$ for all runs. This is suitable for BC because our actions in the range of $[-1, 1]$. For both MCNN+MLP(Fixed) and MCNN+BeT(Fixed), we use the same set of hyperparameters across all tasks. In particular, the MCNN-specific hyperparameter values are $\lambda = 0.1$ and 10% memories. We show an abltaion study of normalized scores with varying number of memories (2.5%-80%) in Figure 14. We also show an ablation study with both varying $\lambda$ (0.1, 1.0, 10.0, 100.0) values and number of memories (2.5%, 5%, and 10%) in the 3D bar charts of Figure 15. We also tabulate the highest value across all MCNN-specific hyperparameters (obtained by evaluating on 20 trajectories) in the MCNN+MLP(Tuned) and MCNN+BeT(Tuned) columns of Tables 2 and 3.

**Implementation details for Figure 1:** We run MCNN+MLP experiments on randomly sampled subsets of the human and expert task datasets and report our performance normalized with that of the D4RL BC scores from [9]. This means that for many points in Figure 1 we used a smaller set of datapoints from a particular task for training an MCNN+MLP policy and yet performed better than D4RL BC's score on that task.

**Discussion on computational costs:** The computation cost of MCNN during inference is dominated by the 1 nearest neighbors search among the $M$ memories in an observation space $\mathbb{R}^d$. From the neural gas, we obtain a graph connecting memories. The search for the nearest memory is $O(Md)$ for the first observation. For every subsequent observation, we simply perform a nearby breadth-first search starting from the previous memory. This is $\Theta(\text{nearby search depth} * d)$ in the average case. Please note that the nearby search depth is kept small (1, 2, 3, 4). The search for the first memory can also be further reduced to $O(d \log M)$ with the K-D trees data structure. Alternatively, leveraging the parallel processing power of GPUs, this search can be done very quickly (<1 ms) in practice even without a graph and by simply searching amongst all memories for the closest memory.

Further, since the vanilla neural net is a component of MCNN, we are less efficient at inference than the vanilla neural net component alone. But, MCNN is more efficient than baselines like VINN and 1-NN since we only have $M$ memories, and this value is set to 10% of the total number of datapoints or lower. VINN and 1-NN have to perform nearest neighbors search on a much larger

dataset than MCNN. Further VINN requires the top k neighbors (usually k=10) while we require only the 1 nearest memory.

## G  ADDITIONAL FIGURES AND DETAILED RESULTS TABLES

We tabulate all reward values from our previous bar charts in Table 2. We tabulate all corresponding normalized scores obtained following the methodology in D4RL [9] in Table 3. We plot all the ablation results for replacing neural gas memories with random memories for human and expert tasks in Figure 13. We plot performance variation against number of memories (from 2.5% to 80%) for all human tasks in Figure 14. We also plot a comparison of scores with varying $\lambda$s (0.1, 1.0, 10.0, 100.0) and varying number of memories (2.5%, 5%, and 10%) for each task in the 3D bar charts of Figure 15.

Table 2: Comparison of returns (across 20 evaluation trajectories and 3 random seeds) between baselines, our reimplementation of BC, and our methods – MCNN+MLP, MCNN+BeT, and MCNN+Diff (see smaller table below) on the Adroit and CARLA domains. Here, we show both our MCNN results with the same fixed set of hyperparameters across all tasks (Fixed) and with hyperparameters tuned online (Tuned).

| Task Name | Baselines | | | | | | | Ours | | | |
| --- | --- | --- | --- | --- | --- | --- | --- | --- | --- | --- | --- |
| | D4RL BC [9] | BeT-BC [38] | 1-NN | VINN [26] | CQL-Sparse [18] + 0/1 reward | Implicit BC [7] | MLP-BC (Reimpl.) | MCNN + MLP (Fixed Hypers) | MCNN + MLP (Tuned Hypers) | MCNN + BeT (Fixed Hypers) | MCNN + BeT (Tuned Hypers) |
| pen-human-v1 | 1122 | 841 ± 60 | 1982 ± 227 | 1490 ± 152 | 377 ± 55 | 2586 ± 65 | 1845 ± 213 | 3285 ± 209 | **3405** ± 328 | 1050 ± 119 | 1089 ± 110 |
| pen-expert-v1 | 2633 | 1853 ± 117 | 3102 ± 275 | 3157 ± 88 | 671 ± 14 | - | 3194 ± 127 | 3947 ± 227 | **4051** ± 195 | 2033 ± 1.8 | 2103 ± 101 |
| pen-cloned-v1 | 1792 | 1348 ± 28 | 1902 ± 148 | 1909 ± 35 | - | - | 1806 ± 72 | 2208 ± 82 | **2820** ± 119 | 1595 ± 15 | 1604 ± 12 |
| hammer-human-v1 | -79 | -189 ± 11 | -66 ± 20 | -232 ± 10 | -241 ± 0.3 | -132 ± 25 | -11 ± 118 | 262 ± 107 | **262** ± 107 | -130 ± 52 | -130 ± 52 |
| hammer-expert-v1 | 16140 | 2731 ± 261 | 10069 ± 770 | 10551 ± 1010 | 11311 ± 502 | - | 13710 ± 2002 | 16027 ± 383 | **16387** ± 392 | 3605 ± 663 | 4417 ± 297 |
| hammer-cloned-v1 | -170 | -235 ± 3.9 | -207 ± 17 | -230 ± 7.8 | - | - | -232 ± 13 | -233 ± 5.2 | **-155** ± 61 | -229 ± 2.6 | -228 ± 1.3 |
| relocate-human-v1 | -6.4 | -2.7 ± 1.6 | -4.7 ± 0.4 | -4.7 ± 0.0 | -20 ± 0.4 | -0.1 ± 2.1 | -5.2 ± 1.7 | -4.7 ± 0.8 | -4.7 ± 0.8 | 8.2 ± 4.2 | **10** ± 6.8 |
| relocate-expert-v1 | 4289 | 490 ± 42 | 1095 ± 268 | 1283 ± 123 | 1910 ± 1168 | - | 4361 ± 55 | 4566 ± 47 | **4566** ± 47 | 558 ± 66 | 558 ± 66 |
| relocate-cloned-v1 | -11 | -4.9 ± 0.1 | -8.5 ± 2.1 | -9.0 ± 0.4 | - | - | -9.0 ± 0.4 | -8.1 ± 0.4 | -6.0 ± 0.4 | -2.9 ± 2.1 | **-0.3** ± 1.3 |
| door-human-v1 | -42 | -54 ± 0.1 | -25 ± 8.5 | -39 ± 9.4 | -66 ± 0.3 | **361** ± 67 | -53 ± 2.3 | -5.4 ± 15 | 9.0 ± 29 | -54 ± 0.3 | -53 ± 0.3 |
| door-expert-v1 | 969 | 560 ± 256 | 2716 ± 24 | 2760 ± 2.6 | 2731 ± 117 | - | 2798 ± 33 | 3033 ± 0.3 | **3035** ± 7.0 | 902 ± 50 | 1038 ± 141 |
| door-cloned-v1 | -59 | -59 ± 0.2 | -59 ± 0.6 | -59 ± 0.0 | - | - | -59 ± 0.3 | -59 ± 0.3 | -59 ± 0.9 | **-58** ± 0.1 | **-58** ± 0.1 |
| carla-lane-v0 | 325 | 327 ± 14 | 348 ± 30 | 297 ± 41 | - | - | 358 ± 31 | 441 ± 35 | **466** ± 50 | 390 ± 2.7 | 390 ± 2.7 |
| carla-town-v0 | **-161** | - | -458 ± 179 | -315 ± 102 | - | - | -497 ± 128 | -511 ± 210 | -465 ± 215 | - | - |

| Task Name | Baseline | Ours | |
| --- | --- | --- | --- |
| | Diff-BC [44] | MCNN + Diff (Fixed Hypers) | MCNN + Diff (Tuned Hypers) |
| pen-human-v1 | 2021.41 ± 46.50 | 2188.03 ± 14.01 | 2345.40 ± 160.95 |
| pen-expert-v1 | 3194.86 ± 424.14 | 3516.77 ± 284.64 | 3568.63 ± 284.05 |
| pen-cloned-v1 | - | - | - |
| hammer-human-v1 | -159.85 ± 56.20 | 17.89 ± 128.08 | 130.28 ± 154.21 |
| hammer-expert-v1 | 14044.84 ± 121.54 | 16088.83 ± 67.96 | 16181.62 ± 88.87 |
| hammer-cloned-v1 | - | - | - |
| relocate-human-v1 | -4.31 ± 1.70 | -2.61 ± 1.70 | 0.78 ± 3.82 |
| relocate-expert-v1 | 4361.09 ± 12.72 | 4572.25 ± 15.27 | 4574.80 ± 16.96 |
| relocate-cloned-v1 | - | - | - |
| door-human-v1 | 121.77 ± 44.94 | 179.04 ± 57.27 | 197.25 ± 32.60 |
| door-expert-v1 | 2800.39 ± 11.16 | 3000.40 ± 3.23 | 3032.12 ± 1.47 |
| door-cloned-v1 | - | - | - |
| carla-lane-v0 | - | - | - |
| carla-town-v0 | - | - | - |

Table 3: Comparison of normalized scores (across 20 evaluation trajectories and 3 random seeds) between baselines, our reimplementation of BC, and our methods – MCNN+MLP, MCNN+BeT, and MCNN+Diff (see smaller table below), on the Adroit and CARLA domains. Here, we show both our MCNN results with the same fixed set of hyperparameters across all tasks (Fixed) and with hyperparameters tuned online (Tuned).

| Task Name | Baselines | | | | | | | Ours | | | |
|---|---|---|---|---|---|---|---|---|---|---|---|
| | D4RL BC [9] | BeT-BC [38] | 1-NN | VINN [26] | CQL-Sparse [18] + 0/1 reward | Implicit BC [7] | MLP-BC (ReImpl.) | MCNN + MLP (Fixed Hypers) | MCNN + MLP (Tuned Hypers) | MCNN + BeT (Fixed Hypers) | MCNN + BeT (Tuned Hypers) |
| pen-human-v1 | 34.4 | 25.0 ± 2.0 | 63.27 ± 7.63 | 46.77 ± 5.10 | 9.43 ± 1.86 | 83.53 ± 2.18 | 58.68 ± 7.14 | 107.0 ± 7.00 | 111.01 ± 11.00 | 32.0 ± 4.0 | 33.3 ± 3.7 |
| pen-expert-v1 | 85.1 | 58.95 ± 3.91 | 100.83 ± 9.23 | 102.68 ± 2.94 | 19.28 ± 0.48 | - | 103.94 ± 4.25 | 129.20 ± 7.63 | 132.70 ± 6.55 | 64.98 ± 0.06 | 67.34 ± 3.39 |
| pen-cloned-v1 | 56.9 | 42 ± 0.94 | 60.60 ± 4.96 | 60.81 ± 1.18 | - | - | 57.36 ± 2.43 | 70.86 ± 2.75 | 91.38 ± 4.00 | 50.3 ± 0.52 | 50.6 ± 0.4 |
| hammer-human-v1 | 1.5 | 0.66 ± 0.087 | 1.60 ± 0.15 | 0.33 ± 0.08 | 0.26 ± 0.002 | 1.09 ± 0.19 | 2.02 ± 0.9 | 4.11 ± 0.82 | 4.11 ± 0.82 | 1.11 ± 0.4 | 1.11 ± 0.4 |
| hammer-expert-v1 | 125.6 | 23.0 ± 2.0 | 79.15 ± 5.89 | 82.84 ± 7.73 | 88.65 ± 3.84 | - | 107.01 ± 15.32 | 124.74 ± 2.93 | 127.49 ± 3.00 | 29.69 ± 5.07 | 35.9 ± 2.27 |
| hammer-cloned-v1 | 0.8 | 0.302 ± 0.03 | 0.52 ± 0.13 | 0.34 ± 0.06 | - | - | 0.33 ± 0.10 | 0.32 ± 0.04 | 0.92 ± 0.47 | 0.353 ± 0.02 | 0.36 ± 0.01 |
| relocate-human-v1 | 0.0 | 0.089 ± 0.038 | 0.04 ± 0.01 | 0.04 ± 0.00 | -0.315 ± 0.01 | 0.15 ± 0.05 | 0.03 ± 0.04 | 0.04 ± 0.02 | 0.04 ± 0.02 | 0.345 ± 0.10 | **0.394** ± 0.16 |
| relocate-expert-v1 | 101.3 | 11.7 ± 1. | 25.97 ± 6.33 | 30.40 ± 2.89 | 45.19 ± 27.54 | - | 103. ± 1.3 | 107.84 ± 1.10 | **107.84** ± 1.10 | 13.32 ± 1.56 | 13.32 ± 1.56 |
| relocate-cloned-v1 | -0.1 | 0.036 ± 0.003 | -0.05 ± 0.05 | -0.06 ± 0.01 | - | - | -0.06 ± 0.01 | -0.04 ± 0.01 | 0.01 ± 0.01 | 0.084 ± 0.05 | **0.145** ± 0.03 |
| door-human-v1 | 0.5 | 0.096 ± 0.002 | 1.07 ± 0.29 | 0.61 ± 0.32 | -0.336 ± 0.01 | **14.22** ± 2.28 | 0.13 ± 0.08 | 1.74 ± 0.51 | 2.23 ± 1.00 | 0.098 ± 0.01 | 0.11 ± 0.01 |
| door-expert-v1 | 34.9 | 20.98 ± 8.7 | 94.39 ± 0.81 | 95.88 ± 0.09 | 94.9 ± 4.0 | - | 97.2 ± 1.13 | 105.18 ± 0.01 | **105.26** ± 0.24 | 32.62 ± 1.7 | 37.27 ± 4.8 |
| door-cloned-v1 | -0.1 | -0.068 ± 0.007 | -0.08 ± 0.02 | -0.07 ± 0.00 | - | - | -0.10 ± 0.01 | -0.10 ± 0.01 | -0.07 ± 0.03 | -0.05 ± 0.005 | **-0.05** ± 0.005 |
| carla-lane-v0 | 31.8 | 32.0 ± 1.4 | 34.03 ± 2.95 | 29.09 ± 3.96 | - | - | 35.03 ± 3.00 | 43.14 ± 3.40 | 45.59 ± 4.90 | 38.13 ± 0.26 | 38.2 ± 0.26 |
| carla-town-v0 | **-1.8** | - | -13.45 ± 7.00 | -7.85 ± 4.00 | - | - | -14.95 ± 5.00 | -15.52 ± 8.20 | -13.70 ± 8.40 | - | - |

| Task Name | Baseline | Ours | |
|---|---|---|---|
| | Diff-BC [44] | MCNN + Diff (Fixed Hypers) | MCNN + Diff (Tuned Hypers) |
| pen-human-v1 | 64.59 ± 1.56 | 70.18 ± 0.47 | 75.46 ± 5.4 |
| pen-expert-v1 | 103.96 ± 14.23 | 114.76 ± 9.55 | 116.5 ± 9.53 |
| pen-cloned-v1 | - | - | - |
| hammer-human-v1 | 0.88 ± 0.43 | 2.24 ± 0.98 | 3.1 ± 1.18 |
| hammer-expert-v1 | 109.57 ± 0.93 | 125.21 ± 0.52 | 125.92 ± 0.68 |
| hammer-cloned-v1 | - | - | - |
| relocate-human-v1 | 0.05 ± 0.04 | 0.09 ± 0.04 | 0.17 ± 0.09 |
| relocate-expert-v1 | 103.0 ± 0.3 | 107.98 ± 0.36 | 108.04 ± 0.4 |
| relocate-cloned-v1 | - | - | - |
| door-human-v1 | 6.07 ± 1.53 | 8.02 ± 1.95 | 8.64 ± 1.11 |
| door-expert-v1 | 97.27 ± 0.38 | 104.08 ± 0.11 | 105.16 ± 0.05 |
| door-cloned-v1 | - | - | - |
| carla-lane-v0 | - | - | - |
| carla-town-v0 | - | - | - |

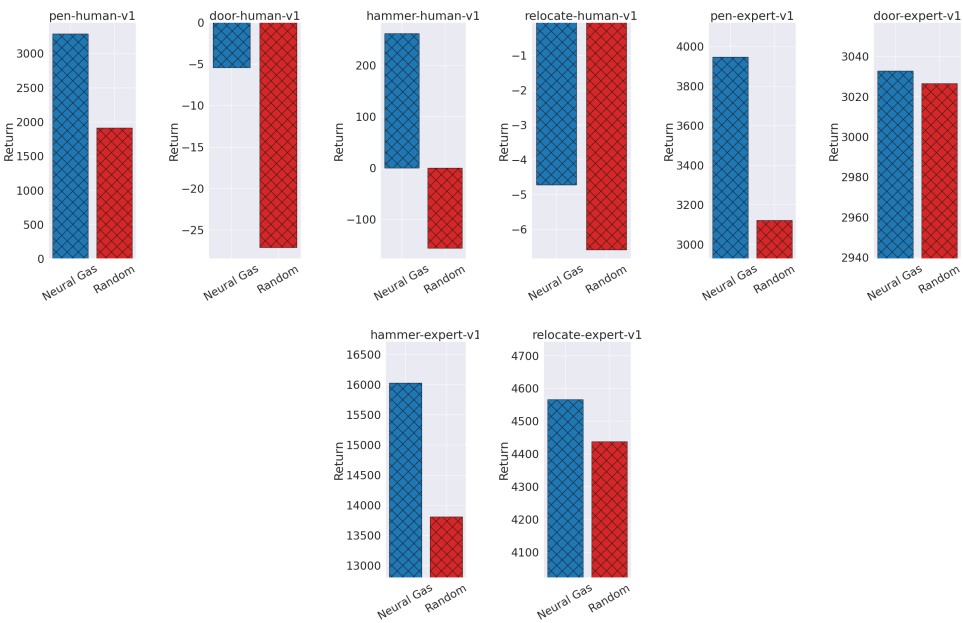

Figure 13: Neural Gas vs Random Memories in MCNN+MLP: Comparison of returns (across 20 eval trajectories) between our method with neural-gas-based memories and randomly chosen memories. This shows that MCNN+MLP performs better with neural gas memories. We attribute this to the spread-out nature of neural gas memories that reduces the distance to the most isolated state, improving imitation performance.

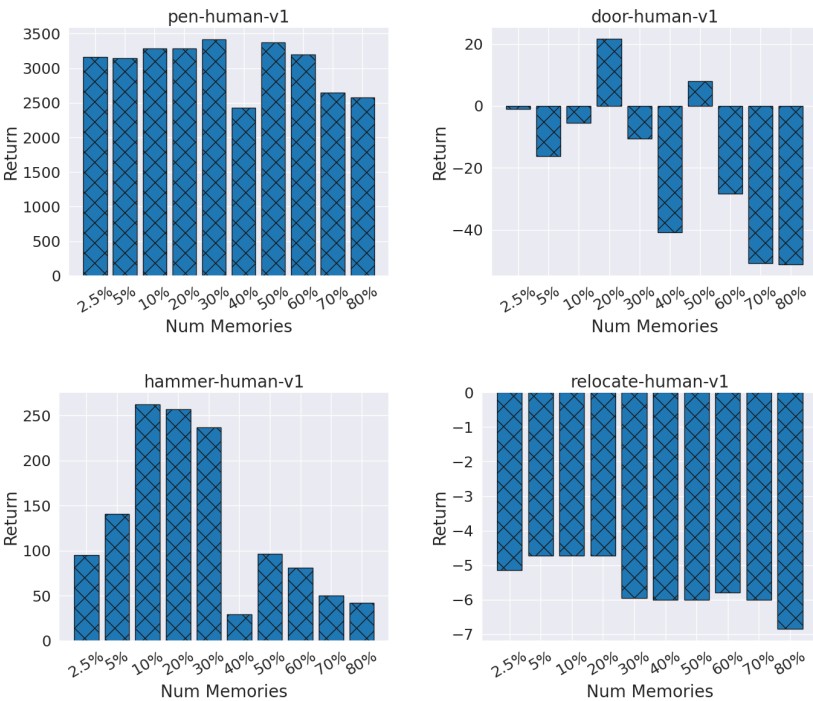

Figure 14: Number of Memories in MCNN+MLP: Comparison of returns (across 20 evaluation trajectories) with our method (MCNN+MLP) using 2.5% of the dataset as memories up to 80% of the dataset as memories. We find a "sweet spot" for num memories at 10-20%. We also see the expected decrease to 1-NN performance with as num memories increases to 100%.

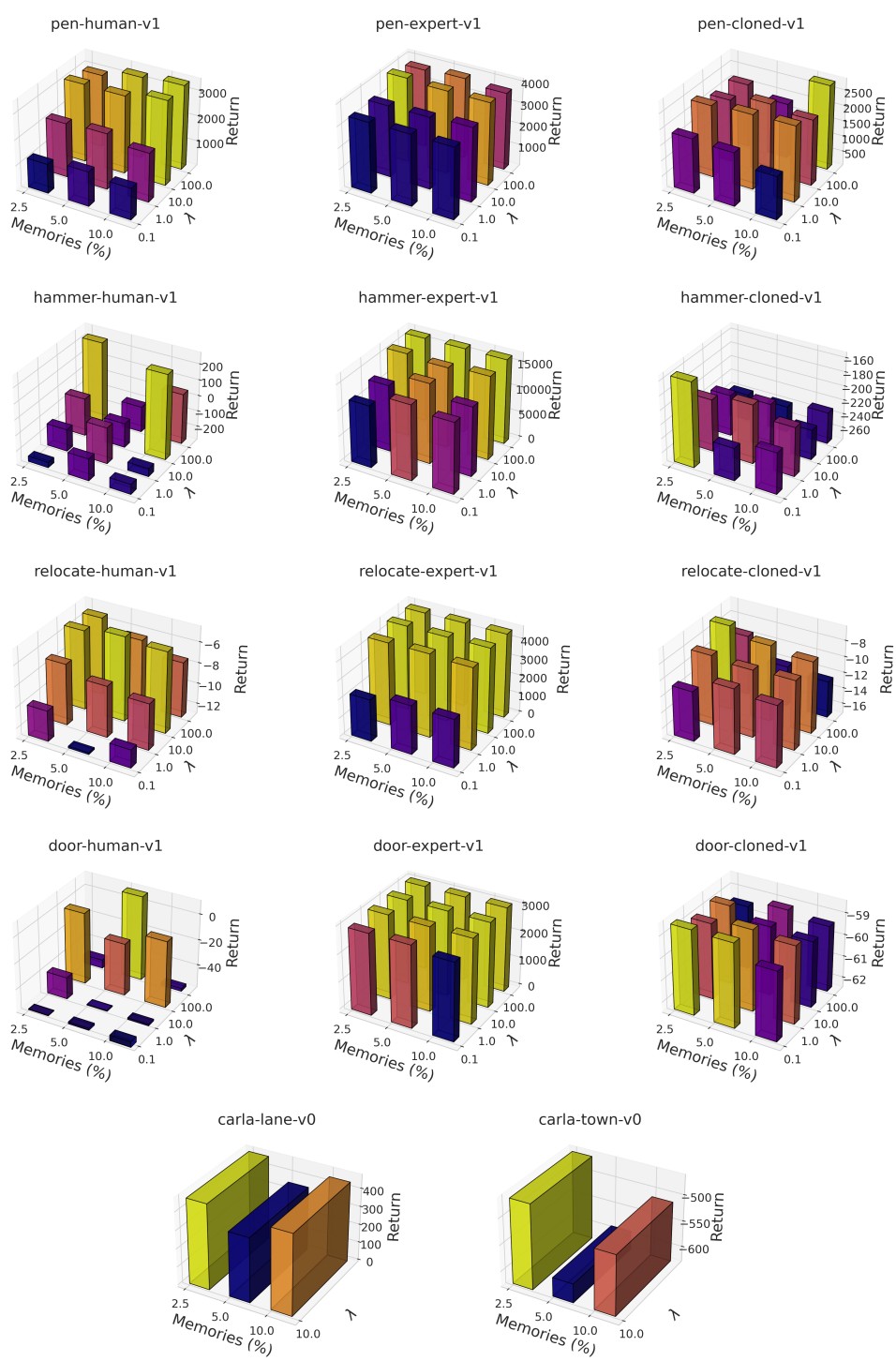

Figure 15: Bar chart of returns for our method against various combinations of $\lambda$ and memories for each task. Each of the 12 bars in each of the 14 subfigures represents the average performance across 20 evaluation trajectories and three seeds. We notice that the best performance can be obtained for $\lambda$ values at the middle, *i.e.*, $\lambda \in \{0.1, 1.0\}$, for any number of memories. This is where our method can interpolate, by design, between the nearest memories and vanilla BC. These plots also demonstrate that by training MCNNs on offline data for a few sets of hyperparameters and simply choosing the best hyperparameter with limited online interaction, we can obtain significant improvements in performance.

**Figure 1 values:** We provide the mean and standard deviation of the return values for various number of demos for various tasks (across various runs) that were in the scatter plot in Figure 1 in Tables 4 and 5. Other points plotted in Figure 1 include the return values (across various runs) for all 9 tasks at the full dataset sizes (25 demos for Adroit human tasks, 5000 for Adroit expert tasks, 400 for the CARLA task) as given in Table 2.

Table 4: Returns with various methods on **pen-expert-v1 and door-expert-v1** for subsampled and full datasets (where NA = Not Available). These values represent the means and standard deviations obtained from various points (corresponding to various runs) in the scatter plot in Figure 1.

| Task | Method | Number of demonstrations | | | | | |
|------|--------|-----------|-----------|------------|------------|------------|-------------------------|
| | | 100 demos | 500 demos | 1000 demos | 2000 demos | 4000 demos | 5000 demos (full dataset) |
| pen-expert-v1 | MCNN+MLP | 2724.61 $\pm$ 138.82 | 2931.19 $\pm$ 31.84 | 2991.88 $\pm$ 18.31 | 3016.65 $\pm$ 9.83 | 3025.47 $\pm$ 3.31 | 3035 $\pm$ 7.0 |
| | D4RL BC | NA | | | | | 969 |
| door-expert-v1 | MCNN+MLP | NA | 3711.79 $\pm$ 31.6 | 3807.97 $\pm$ 5.58 | 3857.74 $\pm$ 29.16 | 3933.75 $\pm$ 41.83 | 4051 $\pm$ 195 |
| | D4RL BC | NA | | | | | 2633 |

Table 5: Returns with various methods on **pen-human-v1 and hammer-human-v1** for subsampled and full datasets (where NA = Not Available). These values represent the means and standard deviations obtained from various points (corresponding to various runs) in the scatter plot in Figure 1.

| Task | Method | Number of demonstrations | | | | |
|------|--------|---------|----------|----------|----------|------------------------|
| | | 5 demos | 10 demos | 15 demos | 20 demos | 25 demos (full dataset) |
| pen-human-v1 | MCNN+MLP | 2169.45 $\pm$ 189.76 | 2542.62 $\pm$ 128.38 | 2673.36 $\pm$ 100.93 | 2988.21 $\pm$ 155.0 | 3405 $\pm$ 328 |
| | D4RL BC | NA | | | | 1122 |
| hammer-human-v1 | MCNN+MLP | 25.29 $\pm$ 5.57 | 36.19 $\pm$ 3.27 | 113.73 $\pm$ 3.08 | 154.24 $\pm$ 8.01 | 253.57 $\pm$ 3.26 |
| | D4RL BC | NA | | | | -79 |

**Oversampling memories:** We also compare MCNN+MLP with a vanilla MLP-BC algorithm that oversamples the memories which comprise 10% of the dataset and report results in Table 6. It can be clearly seen that MCNN+MLP significantly outperforms the oversampled MLP-BC for various amounts of oversampling. Moreover, MLP+BC with oversampling performs similar to vanilla MLP+BC.

Table 6: Comparison of MCNN+MLP with vanilla MLP-BC with memories (or codebook entries) oversampled by various amounts. Here, we use '$N$x sampled' to denote that each memory was sampled $N$ times in each training epoch.

| Task Name | Baselines | | | | Ours | |
|-----------|-----------|-----------|-----------|-----------|--------------------|--------------------|
| | MLP-BC | | | | MCNN + MLP | MCNN + MLP |
| | normal | 2x sampled | 3x sampled | 6x sampled | (Fixed Hypers) | (Tuned Hypers) |
| pen-human-v1 | 1845 $\pm$ 213 | 2080.13 | 2093.24 | 2114.11 | 3285 $\pm$ 209 | **3405** $\pm$ 328 |
| pen-expert-v1 | 3194 $\pm$ 127 | 3106.64 | 3136.44 | 3285.47 | 3947 $\pm$ 227 | **4051** $\pm$ 195 |
| hammer-human-v1 | -11 $\pm$ 118 | -10.86 | -10.86 | -4.33 | 262 $\pm$ 107 | **262** $\pm$ 107 |
| hammer-expert-v1 | 13710 $\pm$ 2002 | 13447.58 | 13447.58 | 13970.34 | 16027 $\pm$ 383 | **16387** $\pm$ 392 |
| relocate-human-v1 | -5.2 $\pm$ 1.7 | -5.95 | -5.94 | -5.66 | -4.7 $\pm$ 0.8 | **-4.7** $\pm$ 0.8 |
| relocate-expert-v1 | 4361 $\pm$ 55 | 4403.49 | 4403.49 | 4445.89 | 4566 $\pm$ 47 | **4566** $\pm$ 47 |
| door-human-v1 | -53 $\pm$ 23 | -48.64 | -46.79 | -35.6 | -5.4 $\pm$ 15 | **9.0** $\pm$ 29 |
| door-expert-v1 | 2798 $\pm$ 33 | 2821.83 | 2851.2 | 2909.94 | 3033 $\pm$ 0.3 | **3035** $\pm$ 7 |

