# OpenReview forum: "Memory-Consistent Neural Networks for Imitation Learning"
_ICLR.cc/2024/Conference — ICLR 2024 poster_

### Official Review · Reviewer_L3an · 2023-10-29

**Soundness:** 3 good
**Presentation:** 3 good
**Contribution:** 2 fair
**Rating:** 6
**Confidence:** 4

**Summary:**

The paper proposes a mechanism to cope with the loss of alignment between training and online evaluation in behavior cloning. It first collects state-action pairs from training datasets and then combines the nearest-neighbor state outputs with regular neural network predicted output, such that the collected output can be chosen for observed states identical to existing ones in the training dataset and the predicted neural network output can be chosen for novel states. The results are based off of 9 tasks across 5 environments and compared to 7 baselines.

**Strengths:**

- I appreciate the effort to formalize most of the definitions included in the paper.
- The experiments contain a good amount of baselines and model variants.

**Weaknesses:**

- The proposed method seems equivalent to supervised training of a NN with the codebook entries oversampled. (Edit: this has been clarified by the authors in an additional experiment)
- It is unclear to me why the clipping function in Fig. 12 should be referred to as "tanh-like".
- The proposed method seems to quickly drop the performance gains as the number of training data increases, which might render it irrelevant for tasks requiring a regular amount of training examples. (Edit: performance in low-data regime seems actually competitive upon revisiting the results)

**Questions:**

- In Fig. 1, what are the absolute returns for ~10, ~100, ~1,000 and ~10,000 trajectories?

---

> ### Author Response · Authors · 2023-11-16
> **Official Comment by Authors**
>
> We thank the reviewer for their efforts. Please let us know if you have lingering questions and whether we can provide any additional clarifications during the discussion period to improve your rating of our paper.
>
> > The proposed method seems equivalent to supervised training of a NN with the codebook entries oversampled.
>
> We respectfully disagree and note that the MCNN is not equivalent to a neural network with memories oversampled. For all the training data points that are not memories (upto 97.5% of the dataset when memories are 2.5%), the nearest neighbor component and the neural network (both weighted by their corresponding exponential distance factor) contribute to the predicted action. This is particularly effective as seen in Figures 4, 5, and 8 across various tasks, environments, demonstration data, input modalities, and neural network architectures in beating 7 state-of-the-art baselines including various vanilla neural network architectures, all of which have access to the exact same dataset.
>
> We performed a comparison between MCNN+MLP and MLP-BC with memories oversampled by various amounts and reported these **new results in Appendix F (Table 5)**. Across all Adroit tasks, we see that MCNN+MLP significantly outperforms the oversampled MLP-BC.
>
> We also respectfully note that other reviewers disagree. Reviewer Cf33 describes our method as “an interesting approach for tackling behavior cloning. Even more so since this can be combined with underlying network architectures”, reviewer kyQU describes our method as “simple yet elegant” and reviewer 5EEF describes our method as providing “something that does not exist for vanilla neural networks”.
>
> > It is unclear to me why the clipping function in Fig. 12 should be referred to as "tanh-like".
>
> We thank the reviewer for the question. We describe it as tanh-like since it maps inputs to a value in [-1, 1] just like the familiar tanh activation function in the deep learning literature. But, it is actually a combination of ReLUs as given in Algorithm 2, detailed in Appendix E, and depicted in Figure 12.
>
> > The proposed method seems to quickly drop the performance gains as the number of training data increases, which might render it irrelevant for tasks requiring a regular amount of training examples.
>
> We believe the reviewer is referring to Figure 1. In Figure 1, we plot $\frac{| \text{MCNN return with subset of data} - \text{D4RL BC return with full data} |}{| \text{D4RL BC return with full data} |}$ for various tasks. This means that we improve over D4RL BC’s performance even when using fewer demonstrations than D4RL BC by upto 200+% on various tasks. For a large number of demonstrations, we still have a significant median improvement of around 50% with many points above a large 200% improvement. We briefly described this in Appendix E under “Implementation details for Figure 1”.
>
> Moreover, Figure 1 aims to highlight that, with few demonstrations, the performance gains are even more impressive with a median above 150% improvement. This is notable in imitation learning because, usually, only a small number of human demos are available in the real world [1, 2, 3, 4]. Many recent papers in robot learning emphasize the importance of learning from few demonstrations [1, 2, 3, 4]. Generalizing from these few demos to test environments is particularly challenging [1, 2, 3, 4]. We show that our method overcomes this challenge and excels in this low-data regime.
>
> Moreover, on a particular task, more data (as expected) leads to better performance. We can see this in the **newly added Tables 3 and 4 in Appendix F**. As requested by the reviewer, in these two new tables, we provide the mean and standard deviation of the return values for various number of demos for various tasks (across various runs) that were in the scatter plot in Figure 1.
>
> [1] Haldar et al, Teach a Robot to FISH: Versatile Imitation from One Minute of Demonstrations, 2023.
>
> [2] Haldar et al, Watch and match: Supercharging imitation with regularized optimal transport, 2023.
>
> [3] Arunachalam et al, Dexterous Imitation Made Easy: A Learning-Based Framework for Efficient Dexterous Manipulation, 2023.
>
> [4] Kostrikov et al, Discriminator-actor-critic: Addressing sample inefficiency and reward bias in adversarial imitation learning, 2019.
>
> > In Fig. 1, what are the absolute returns for ~10, ~100, ~1,000 and ~10,000 trajectories?
>
> As requested by the reviewer, in the **newly added Tables 3 and 4 in Appendix F**, we provide the mean and standard deviation of the return values for various number of demos for various tasks (across various runs) that were in the scatter plot in Figure 1.

---

> > ### Comment · Reviewer_L3an · 2023-11-16
> > **Thank you for the effort**
> >
> > I thank the authors for the additional experiments, especially for comparing to oversampling codebook entries during training (which also improves, even if less so than I had expected, over the baseline). About Fig. 1, I was afraid the denominator was from a baseline with the corresponding data-subset regime, but I'm glad that's not the case. Given the new evidence provided by the authors (I still disagree that a clipping function should be named "tanh-like", even if their values match asymptotically and both cross zero at the origin), I am happy to adjust my score.

---

> > > ### Author Response · Authors · 2023-11-16
> > > **Thank you**
> > >
> > > Thank you for the quick response and for updating your score. We are happy to paraphrase the “tanh-like” description in the next version.

---

### Official Review · Reviewer_5EEF · 2023-10-30

**Soundness:** 3 good
**Presentation:** 3 good
**Contribution:** 3 good
**Rating:** 6
**Confidence:** 3

**Summary:**

The paper proposes a semi-parametric method for behavior cloning, MCNN,  that combines non-parametric nearest neighbor based policy learning with paramteric neural network based policies. As a result of this amalgam, the authors show that the MCNN class of functions are bounded in width and the suboptimality gap, something that does not exist for vanilla neural networks. Further, the authors provide results across 5 environments where adding MCNN to a new architecture consistently improves the results.

**Strengths:**

- The paper proposes a new class of functions that combines non-parametric nearest neighbor based policy learning with parametric neural network based policies. As a result of this amalgam, the authors show that the MCNN class of functions is bounded in width and the suboptimality gap, something that does not exist for vanilla neural networks.
- The authors provide results across 5 environments where adding MCNN to a new architecture consistently improves the results.
- The authors ablate the performance of MCNN across a different number of memories and the optimal number of memories is significantly smaller than the entire dataset. This highlights the efficiency of the non-parametric portion of the algorithm as compared to approaches like VINN and 1-NN.
- The proposed method also seems to work on image inputs from the CARLA environment.

**Weaknesses:**

- The authors mention that they provide results on 9 tasks across 5 environments. But I only see 5 tasks, 1 per environment. It would be great if the authors could clarify the 9 tasks that they evaluate on and where they have provided the results.
- For CARLA, the images have been embedded using a fixed off-the-shelf ResNet34 encoder. This might not be ideal for more complicated visual scenes such as the Franka Kitchen environment used in BeT and Diffusion Policy. It would be great if the authors could evaluate MCNN in this benchmark and provide some comments on possible pretraining approaches for the encoder when such an off-the-shelf encoder is insufficient.
- *“ MCNN can even improve the performance of simple MLP architectures to beyond that of more sophisticated recent architectures such as Diffusion models.”*: In quite a few cases, MCNN works better with MLP than with BeT or Diffusion Policy. Both BeT and Diffusion Policy were developed to deal with multimodal data distributions. Hence, I wonder if the tasks shown are not multimodal enough. Evaluating MCNN on the Franka Kitchen environment used in BeT and Diffusion Policy would help clarify this.

**Questions:**

It would be great if the authors could address the points mentioned in “Weaknesses”.

---

> ### Author Response · Authors · 2023-11-16
> **Official Comment by Authors (Part 1 of 2)**
>
> We thank the reviewer for their detailed and thorough review and pertinent questions. Please let us know if you have lingering questions and whether we can provide any additional clarifications during the discussion period to improve your rating of our paper.
>
> > For CARLA, the images have been embedded using a fixed off-the-shelf ResNet34 encoder. This might not be ideal for more complicated visual scenes such as the Franka Kitchen environment used in BeT and Diffusion Policy. It would be great if the authors could evaluate MCNN in this benchmark and provide some comments on possible pretraining approaches for the encoder when such an off-the-shelf encoder is insufficient.
>
> > “MCNN can even improve the performance of simple MLP architectures to beyond that of more sophisticated recent architectures such as Diffusion models.”: In quite a few cases, MCNN works better with MLP than with BeT or Diffusion Policy. Both BeT and Diffusion Policy were developed to deal with multimodal data distributions. Hence, I wonder if the tasks shown are not multimodal enough. Evaluating MCNN on the Franka Kitchen environment used in BeT and Diffusion Policy would help clarify this.
>
> Thank you very much. As the reviewer requested, we have evaluated MCNN, and in particular MCNN+Diffusion, on the Franka Kitchen environment and have **added these new results to Appendix F (Table 6)**. We would like to clarify that the Franka Kitchen environment used in the BeT and Diffusion Policy papers is state/proprioception based. They do not use image inputs. We obtain performance better than (or equal to) diffusion policy in all metrics from success rates in interacting with 1 object up to 5 objects. For 1 to 4 object interaction, diffusion policy is already a strong baseline achieving close to 100%. Our method (MCNN+Diffusion) achieves 100% in all four metrics. For interacting with 5 objects, we see a 4x improvement with MCNN+diffusion over the performance of the best baseline which is diffusion policy.
>
> Also, as requested by the reviewer, we recommend end-to-end training both the vision encoder and policy with the behavior cloning loss when an independently trained vision encoder such as [1,2,3,4] is not sufficient.
>
> We would also like to refer the reviewer to videos of our policies in both Adroit and Franka Kitchen environments and videos of demonstrations in Adroit human tasks available at our anonymous website https://sites.google.com/view/mcnn-anon . The multimodality in Adroit, particularly in the human datasets collected by humans wearing VR gloves, is visible in these videos. Some examples include (1) in the hammer task, some demonstrations move the hammer above the bolt before hitting it, some move it below, and some directly hit the bolt and (2) in the pen task, some demonstrations have the little finger above the pen and some have it below the pen throughout the episode.
>
> Finally, through the statement:
>
> *“MCNN can even improve the performance of simple MLP architectures to beyond that of more sophisticated recent architectures such as Diffusion models”*
>
> we only aimed to indicate that MCNN+MLP outperforms plain Diffusion-BC and not that MCNN+Diffusion did not outperform MCNN+MLP.  Indeed, as seen in Figures 3 and 4, in the majority of the tasks (5 of 8), MCNN+Diffusion outperforms (or equals) MCNN+MLP. Among the baselines too, Diffusion-BC outperforms (or equals) MLP-BC in 7 out of 8 tasks. We note that multimodality is not the only challenge in imitation learning. MCNN helps address the significant compounding error challenge which MLP-BC fails to address and hence results in MCNN+MLP having an overall better performance than Diffusion-BC even though Diffusion is better equipped to handle multimodality. Our results in Franka Kitchen highlight this where MCNN+Diffusion improves upon the already strong Diffusion-BC baseline. We added this clarification to Appendix E.
>
> [1] He et al 2019, Momentum Contrast for Unsupervised Visual Representation Learning
>
> [2] Nair et al 2022, R3M: A Universal Visual Representation for Robot Manipulation
>
> [3] Ma et al 2022, VIP: Towards Universal Visual Reward and Representation
>
> [4] Majumdar et al 2023, Where are we in the search for an Artificial Visual Cortex for Embodied Intelligence?

---

> > ### Author Response · Authors · 2023-11-16
> > **Official Comment by Authors (Part 2 of 2)**
> >
> > > The authors mention that they provide results on 9 tasks across 5 environments. But I only see 5 tasks, 1 per environment. It would be great if the authors could clarify the 9 tasks that they evaluate on and where they have provided the results.
> >
> > Following notation in prior works [1, 2, 3], we define an imitation learning task as a tuple of (environment, reward function, and demonstration dataset). Since we perform experiments with both "expert" and "human" demonstrations, our experiment list has the following 9 tasks: pen-human-v1, pen-expert-v1, hammer-human-v1, hammer-expert-v1, door-human-v1, door-expert-v1, relocate-human-v1, relocate-expert-v1, and carla-lane-v0. We added this clarification to Appendix E.
> >
> > [1] Fu et al, D4RL: Datasets for Deep Data-Driven Reinforcement Learning, 2021.
> >
> > [2] Kumar et al 2020, Conservative Q-Learning for Offline Reinforcement Learning, 2020.
> >
> > [3] Florence et al, Implicit Behavioral Cloning, 2021.

---

> > > ### Author Response · Authors · 2023-11-18
> > > **Have we addressed your concerns?**
> > >
> > > Dear reviewer 5EEF,
> > >
> > > We thank you again for taking the time and effort to help improve our paper. We believe your key concern was on evaluating MCNN in the Franka Kitchen environment. We believe we have addressed your concerns with our newly added results in Appendix F (Table 6) where MCNN+Diffusion outperforms all baselines. Videos of our policy in Franka Kitchen along with our updated code can be found at our anonymous website (https://sites.google.com/view/mcnn-anon).
> > >
> > > We would be grateful for an opportunity to address any pending concerns you can point us to.
> > >
> > > Thank you,
> > >
> > > Authors

---

> ### Author Response · Authors · 2023-11-22
> **Reminder**
>
> Dear reviewer 5EEF,
>
> Since we are at the end of the author-reviewer discussion period, we are again reaching out to ask if our response and new experiments in Franka Kitchen (Table 6, Appendix F) have addressed your concerns. Please let us know if you have lingering questions and whether we can provide any additional clarifications today (the last day of the discussion period) to improve your rating of our paper.
>
> Thank you,
>
> Authors

---

> > ### Comment · Reviewer_5EEF · 2023-11-22
> > **Thank you for the rebuttal**
> >
> > I thank the authors for the comprehensive rebuttal. My concerns have been sufficiently addressed. I am raising my score to 6.

---

> > > ### Author Response · Authors · 2023-11-22
> > > **Thank you**
> > >
> > > Thank you for the response and for updating your score.

---

### Official Review · Reviewer_kyQU · 2023-10-30

**Soundness:** 3 good
**Presentation:** 3 good
**Contribution:** 2 fair
**Rating:** 6
**Confidence:** 4

**Summary:**

In this work, the authors present their work Memory Consistent Neural Networks (MCNN), that aims to improve the performance of traditional behavior cloning techniques by introducing a novel demonstration handling technique, which aims to reduce compounding errors that can commonly occur in the BC framework. The authors augment the BC procedure by incorporating demonstrations provided by the expert, directly into the policy’s decision step. They accomplish this by utilizing a convex combination between the expert’s action for the most similar state encountered in the demonstration set, and the output of a parametrized model approximating the policy. They offer both intuitive and theoretical motivation for their strategy: 1) that past similar experiences can help ground the system’s response to behaviors that have been displayed by an expert and thus avoid drifting into potentially catastrophic state-space areas and 2) offer theoretical guarantees  that indeed the system’s output is constrained within a desired area according to the provided hyperparameter L. MCNN requires that a subset of the available demos be organized in a particular form of memory buffer, referred to as Memory Code Book throughout the work. . Selected demonstrations are structured into a graph G, where each node is connected to its nearest neighbor according to some provided distance function d, defined over the state space, and the edge stands for the action used to transition to that nearest neighbor. They argue that within this framework, only a subset of the training data is required to craft graph G. This graph G, is then used to craft a function F(x) that locates the nearest known state s to x, and outputs the action a that the expert has demonstrated from that state. Using a convex combination between this action a, and the action a’ the policy network predicts for the input state x, the system finally outputs the action a_final as the predicted action for state x. Finally the policy network’s parameters are updated by standard Stochastic Gradient Descent, using a loss signal over the predicted a_final and the expert action for state x.
The method is then compared to several BC based approaches over 5 simulated tasks where reward signals are known and can be used to provide quantitative assessment.

**Strengths:**

The authors present all the material in a very easy to follow manner. All figures are good quality and the results are well displayed.

The authors have performed extensive experimentation and comparison to the baselines. In addition, the baselines selected are reasonable as this work aims to improve the behavior of BC methods. Finally, they offer adequate implementation details in their appendix.

Their method is predicated on a simple yet elegant idea: use expert demonstrations as a human-like memory of experiences to constrain the output of a BC model. In this manner the output is kept within a reasonable distance of the known state space. This, assuming good coverage provided by the expert, can significantly reduce compounding errors that can affect BC techniques (i.e DAGGER). Additionally, the system can potentially better handle not-too-dissimilar unseen states by virtue of projecting them to the known state space via the  nearest neighbor function capability.

Sound theoretical justification for the work’s motivation. Essentially the authors argue that by integrating known state-action  information into the decision process directly they can constrain the output range of the system. They also provide reasoning as to why the code book can only be a subset of the total amount demonstrations, as long as it is crafted in a sophisticated manner. That is, the code book can be small enough to facilitate  fast inference, as long as the selected state-actions are good representatives of the known state-action space.

**Weaknesses:**

The main weakness of this work, as is common with BC approaches, is assumption of representative state-action pairs and optimal behavior provided by the demonstrator. There is no insight as to how the method will behave with reasonably suboptimal demonstrations, as it being a BC based approach has no apparent mechanism that enables it to focus on the better demonstrations of a provided set.
There is recent interest in work that can discern between useful demonstrations and harmful / irrelevant ones that can inhibit training. (i.e TREX, or Subdominance minimization (Ziebart 2022)). While understandably not the direct focus of this work, it can be inhibiting for complex tasks to assume a plethora of very carefully curated demonstrations that are both optimal and extremely descriptive of the entire problem space, hence the aforementioned attempts to augment imitation learning with such capabilities.

In the same direction, there is no experimentation for generalization. For example, with demonstration provided for task A, how would the method perform for a slightly altered task A’ there perhaps the goal has changed slightly, the demonstrations are perturbed by noise or simply slightly different. It is not unreasonable to believe that this method would not perform poorly in such a setting and good results would strengthen the method’s appeal considerably.

Finally, the distance used to select the nearest neighbor, is quite critical for this method and problem specific. For example, the distance needed for an image-based problem could be quite different from a control state-based problem. This can potentially deduct from the appeal of a traditional BC approach that doesn't need such an engineered mechanism. See question 1 and 3 below. (The authors have clarified that the distance used is a generic L2 distance across all problem spaces considered.)

Having revisited the sections pointed out to by the authors, on the parts that I had concerns, namely the generalization aspect and the problem specific distance metric, I believe this work can have utility in the low data data sample regime, and thus I have updated my rating.

**Questions:**

1) What distance measure did you use for your problems? How do you see the performance of your method be impacted from a different measure choice? Would a more generic measure adversely affect performance compared to a more tailored one? I.e Aggregated Manhattan distance for images compared to some kernel-based distance.

2) Relevant to the generalization point raised in the weaknesses section, how do you believe would MCNN behave to inputs based on varying levels of state - perturbation?

3) What was the motivation behind a static nearest neighbor function? Have you considered training it as well? (The authors have clarified that the distance used is a generic L2 distance)

---

> ### Author Response · Authors · 2023-11-16
> **Official Comment by Authors (Part 1 of 2)**
>
> We thank the reviewer for the detailed and thorough review. We appreciate the detailed context provided by the reviewer. Please let us know if you have lingering questions and whether we can provide any additional clarifications during the discussion period to improve your rating of our paper.
>
> > The main weakness of this work, as is common with BC approaches, is assumption of representative state-action pairs and optimal behavior provided by the demonstrator. There is no insight as to how the method will behave with reasonably suboptimal demonstrations, as it being a BC based approach has no apparent mechanism that enables it to focus on the better demonstrations of a provided set. There is recent interest in work that can discern between useful demonstrations and harmful / irrelevant ones that can inhibit training. (i.e TREX, or Subdominance minimization (Ziebart 2022)). While understandably not the direct focus of this work, it can be inhibiting for complex tasks to assume a plethora of very carefully curated demonstrations that are both optimal and extremely descriptive of the entire problem space, hence the aforementioned attempts to augment imitation learning with such capabilities.
>
> > In the same direction, there is no experimentation for generalization. For example, with demonstration provided for task A, how would the method perform for a slightly altered task A’ there perhaps the goal has changed slightly, the demonstrations are perturbed by noise or simply slightly different. It is not unreasonable to believe that this method would not perform poorly in such a setting and good results would strengthen the method’s appeal considerably.
>
> > Relevant to the generalization point raised in the weaknesses section, how do you believe would MCNN behave to inputs based on varying levels of state - perturbation?
>
> Thank you, as you have pointed out, our experimental validation has followed standard practice in the literature on an extensive set of tasks, against well-chosen SOTA baselines. While the reviewer is right that the standard evaluation metrics could be better, we make an earnest effort to closely track the community accepted standards. Among our baselines, nearly all have already proven useful for many realistic tasks with suboptimal demonstrations including on real robots with human-teleoperated demonstrations [1, 2, 3]. Furthermore, note that our experiments already include settings with small, noisy demonstration sets. In all four Adroit human tasks, there are a total of only 25 demonstrations collected by humans wearing VR gloves. As shown in Figure 1, even the lowest number of demonstrations (5 demos) leads to large improvements over the baselines.
>
> We would also like to clarify that all policies in adroit environments are **goal-conditioned**.  The environment provides goal position and orientations as part of the observation vector. Every time that the environment is reset (such as for a new evaluation rollout), the goal is randomly chosen. These include random goal orientations of the pen, goal locations for the relocate task, door locations in door tasks, and nut-and-bolt location in the hammer task. This can be seen clearly in the videos of our policy rollouts at this anonymous link https://sites.google.com/view/mcnn-anon .
>
> With only 25 demonstrations or fewer in Adroit human tasks, it is not possible to cover every random goal location sampled during evaluation rollouts for each of the tasks (such as every possible pen orientation) in the 25 or fewer demos. Hence, coverage too is not available! Even under these stressful conditions, with very few demos that do not provide adequate coverage, MCNN is able to generalize well to new randomly sampled evaluation goals (across 20 evaluation trajectories and 3
> random seeds) and significantly outperform baselines across tasks and environments. We have also added this information to the experimental evaluation section in the paper and in Appendix E.
>
> All this said, we do acknowledge that there is notable space within the imitation learning literature for methods such as T-REX and Subdomain Minimization that filter demonstration datasets. However, we view such improvements as orthogonal to our method which aims at improving the function class.
>
> [1] Chi et al, Diffusion policy: Visuomotor policy learning via action diffusion, 2023.
>
> [2] Pari et al, The surprising effectiveness of representation learning for visual imitation, 2021.
>
> [3] Florence et al, Implicit Behavioral Cloning, 2021.

---

> ### Author Response · Authors · 2023-11-16
> **Official Comment by Authors (Part 2 of 2)**
>
> > Finally, the distance used to select the nearest neighbor, is quite critical for this method and problem specific. For example, the distance needed for an image-based problem could be quite different from a control state-based problem. This can potentially deduct from the appeal of a traditional BC approach that doesn't need such an engineered mechanism.
>
> > What distance measure did you use for your problems? How do you see the performance of your method be impacted from a different measure choice? Would a more generic measure adversely affect performance compared to a more tailored one? I.e Aggregated Manhattan distance for images compared to some kernel-based distance.
>
> Thank you for the questions. We would like to clarify that we did not choose a measure tailored to any environment or task and simply used the general L2 distance everywhere (between two proprioceptive states and between two image embeddings). With this general distance measure, we obtain significant performance improvements across tasks and environments as seen in our results section. We refer the reviewer to Section 5 under “Embedding CARLA images” where we note that the nearest memory neighbor function and neural network operate on the same pre-trained image embeddings or proprioceptive states.
>
> > What was the motivation behind a static nearest neighbor function? Have you considered training it as well?
>
> Thank you for the question. We would also like to clarify that we learn a neural gas (and memories) following the neural gas clustering algorithm [Fritke 1994] which is an unsupervised learning algorithm. Hence, our nearest neighbor function is not static but learned depending on the demonstration dataset. This nearest neighbor function helps us control the deviation that MCNNs can have from the expert-recommended actions. We refer the reviewer to the detailed explanation provided under the "Neural gas" subheading in Section 4.3.

---

> > ### Author Response · Authors · 2023-11-18
> > **Have we addressed your concerns?**
> >
> > Dear reviewer kyQU,
> >
> > We thank you again for taking the time and effort to help improve our paper. We believe your key concern was on the common evaluation procedure for BC approaches in the literature. We believe we have addressed your concerns in the first comment above and also elaborated on our Adroit human setup in Appendix E which has suboptimal demonstrations provided by humans wearing VR gloves, in an environment where the goal is chosen randomly for every evaluation rollout, and with a very small number of demonstrations (5 to 25) that cannot cover every randomly sampled goal across many evaluation rollouts and seeds. We also refer the reviewer to videos of both policy rollouts and demonstration episodes across tasks at our anonymous website (https://sites.google.com/view/mcnn-anon).
> >
> > We would be grateful for an opportunity to address any pending concerns you can point us to.
> >
> > Thank you,
> >
> > Authors

---

> > ### Comment · Reviewer_kyQU · 2023-11-21
> >
> > Thank you for the clarification.

---

> > > ### Author Response · Authors · 2023-11-22
> > > **Thank you**
> > >
> > > Thank you for the response and for updating your score.

---

### Official Review · Reviewer_Cf33 · 2023-11-01

**Soundness:** 3 good
**Presentation:** 3 good
**Contribution:** 3 good
**Rating:** 6
**Confidence:** 4

**Summary:**

This paper proposes a method for behavior cloning to address the i.i.d. assumption violation that occurs under supervised learning algorithms. Specifically, a memory-consistent neural network is proposed in which a set of "memories" are used to constrain the output of the neural network to stay within permissible regions. The proposed method is compared to baseline imitation learning algorithms across a set of standard manipulation and autonomous driving tasks.

**Strengths:**

* Constraining the output of the neural network such that it stays "close" to a specified set of data points is an interesting approach for tackling behavior cloning. Even more so since this can be combined with underlying network architectures.
* The empirical evaluation is fairly thorough, and shows reasonable performance gains across many of the tasks.
* The implementation seems straightforward (although this will incur an additional computational cost during training and inference).

**Weaknesses:**

* As a (semi-)parametric method, the computational cost of training and inference scales with the number of memories. There is a discussion on computational complexity in the appendix, but some analysis on training time would be appreciated here as it's difficult to tell whether this is a significant factor.
* The performance improvement seems sensitive to the underlying network architecture and the task. E.g. in Fig. 4, different models exhibit different levels of improvement for each task. This could make it difficult to use in cases where we can't sample from the environment.
* Similarly, it seems that the clustering method requires you to be able to sample from the entire state space, is this the case? If it is not possible to sample from the environment at will, is it possible to do this clustering purely from data? If the method's performance strongly depends on sampling from the environment, then it would only be fair to compare it to interactive imitation learning methods that also sample from the environment like SQIL, GAIL, etc.

**Questions:**

1) Given the distance lookups that are necessary for finding neighbors, how does this impact the wallclock time for training and inference?
2) It seems like in some tasks the model is very sensitive to the number of memories, e.g. carla-town-v0 in Fig. 15. Do you know why this is the case? It seems unintuitive that 5% would perform worse than both 2.5% and 10%. Do  you think that the number of memories needed to attain sufficient performance is related to task/behavior complexity?
3) It seems like there might be a relationship to critical/important states which exhibit a large difference in the expected reward between the best and worst action, i.e. it is costly to recover from a mistake in a critical state. It may be interesting to do an analysis where you identify the critical states in a task and then ablate them from memory. Do you have any thoughts on whether you can do more intelligent memory selection?

---

> ### Author Response · Authors · 2023-11-16
> **Official Comment by Authors (Part 1 of 2)**
>
> We thank the reviewer for their detailed and thorough review and for supporting our work. Please let us know if you have lingering questions and whether we can provide any additional clarifications during the discussion period to improve your rating of our paper.
>
> > As a (semi-)parametric method, the computational cost of training and inference scales with the number of memories. There is a discussion on computational complexity in the appendix, but some analysis on training time would be appreciated here as it's difficult to tell whether this is a significant factor.
>
> > Given the distance lookups that are necessary for finding neighbors, how does this impact the wallclock time for training and inference?
>
> Thank you for the question. We refer the reviewer to the "discussion on computation costs" in Appendix E where we note that the wall time required is less than 1 millisecond for finding the memory for a single (or even a batch of up to size 1024) of datapoints by parallel search on a RTX 3090 GPU. This time is not a significant factor during inference or training. Forward propagation through the MLP/transformer/diffusion models also similarly takes less than a millisecond. Further, even on our largest datasets of 1 million transitions for the Adroit expert tasks, with 10% of the dataset or 10k datapoints as memories, nearest memory search uses less than half the GPU VRAM used by the neural network model (diffusion, BeT, or larger MLPs). This results in no additional GPU VRAM usage over the amount used by the model and hence no additional usage when compared to the other baselines.
>
> > The performance improvement seems sensitive to the underlying network architecture and the task. E.g. in Fig. 4, different models exhibit different levels of improvement for each task. This could make it difficult to use in cases where we can't sample from the environment.
>
> > Similarly, it seems that the clustering method requires you to be able to sample from the entire state space, is this the case? If it is not possible to sample from the environment at will, is it possible to do this clustering purely from data? If the method's performance strongly depends on sampling from the environment, then it would only be fair to compare it to interactive imitation learning methods that also sample from the environment like SQIL, GAIL, etc.
>
> Thank you for the questions.
> First, we would like to clarify that all the results in the main paper except Fig 7 are obtained with no online interaction, i.e., no sampling from the environment. The clustering method is also purely offline. We apologize that our description in the paper might have left room for possible misinterpretations on this front.
>
> We would also like to highlight that all "MCNN + X (Fixed)" methods refer to MCNN with fixed hyperparameters, not tuned with any online interaction, and applied out of the box to new tasks.
>
> Second, yes, we agree that MCNN yields improvements of varying magnitudes for different tasks and different models, but note that this is not a problem for a practitioner. Indeed, this is generally true for most modeling improvements: for example, in the same Fig 4, IBC outperforms MLP by different margins in different tasks. Rather than consistently sized margins, it is much more important for a candidate modeling improvement to have reliably positive margins. This is very much true in our experiments, as demonstrated tellingly in Fig 1, which aggregates results across many tasks and many base models.

---

> > ### Author Response · Authors · 2023-11-16
> > **Official Comment by Authors (Part 2 of 2)**
> >
> > > It seems like in some tasks the model is very sensitive to the number of memories, e.g. carla-town-v0 in Fig. 15. Do you know why this is the case? It seems unintuitive that 5% would perform worse than both 2.5% and 10%. Do you think that the number of memories needed to attain sufficient performance is related to task/behavior complexity?
> >
> > Thank you for the question. While there is indeed some sensitivity to this parameter, we set it to 10 % in all our experiments in the main paper barring Fig 7 where we explicitly gauge the impact of parameter tuning using online environment interactions. Further, this need not entirely be an empirical choice --- we are guided by the following intuition. The available training dataset must be shared between the nearest neighbors function and the neural network in order for MCNN to work. Since generalization beyond training samples comes from the neural network, we expect it to require more of the training data. This intuition is borne out by the general trend of increasing performance until the number of memories is about 10-20% of the dataset followed by a decrease. It is also difficult to run comprehensive experiments with many seeds, and rare deviations from the general trend on individual tasks may come from the fact that these hyperparameter effects were only tested with 3 seeds per choice. It is statistically likely that there will be minor individual deviations between 3-seed statistics and true trends in a very small fraction of our reported experiments.
> >
> > > It seems like there might be a relationship to critical/important states which exhibit a large difference in the expected reward between the best and worst action, i.e. it is costly to recover from a mistake in a critical state. It may be interesting to do an analysis where you identify the critical states in a task and then ablate them from memory. Do you have any thoughts on whether you can do more intelligent memory selection?
> >
> > We thank the reviewer for this insightful comment. Yes, we agree that it would be helpful to explore smarter strategies for memory selection in future work. We speculate that this would need correlating memory location with 'return to go' like metrics. However, in this paper, we used well-known statistical algorithms to place memories which can capture the distribution of points. It turns out that modifying the function class with such data points can have a big impact on the generalization of the learned policy.

---

> > > ### Author Response · Authors · 2023-11-18
> > > **Have we addressed your concerns?**
> > >
> > > Dear reviewer Cf33,
> > >
> > > We thank you again for taking the time and effort to help improve our paper. We believe your key concerns were (1) on the wall clock time and (2) on online sampling. We believe we have addressed your concerns with our first comment above where (1) we point to our appendix where we measured the time to find a memory to be less than 1 millisecond with parallel search on a GPU and (2) clarify that all the results in the main paper except Fig 7 are obtained with no online interaction and apologize for any possible misunderstanding.
> > >
> > > We would be grateful for an opportunity to address any pending concerns you can point us to.
> > >
> > > Thank you,
> > >
> > > Authors

---

> > > > ### Comment · Reviewer_Cf33 · 2023-11-22
> > > >
> > > > Thanks, I appreciate the response. I think this clears up many of my questions, particularly my concern regarding a need for sampling from the environment.

---

> > > > > ### Author Response · Authors · 2023-11-22
> > > > > **Thank you**
> > > > >
> > > > > Thank you for the response.

---

### Author Response · Authors · 2023-11-16
**Official Global Comment by Authors**

Dear reviewers,

We thank you all for your insightful feedback and helpful suggestions.

We are grateful that reviewers Cf33, kqQU, and 5EEF rated our soundness and presentation as good (3). We are happy that reviewers Cf33 and 5EEF also rated our contribution as good (3) and reviewer kqQU described our contribution as “simple yet elegant”. We are thankful that reviewers used the following to describe our work: “interesting approach for tackling behavior cloning. Even more so since this can be combined with underlying network architectures.”, “evaluation is fairly thorough”, “very easy to follow”, “extensive experimentation and comparison to the baselines”, “simple yet elegant”, “Sound theoretical justification”, “shows…something that does not exist for vanilla neural networks”, and “highlights the efficiency of the non-parametric portion of the algorithm”.

We have addressed each weakness and question for all reviewers below their review. We have also made the following changes to the paper and highlighted all changes in red in the pdf.
1. New results on Franka Kitchen in Appendix F (Table 6) as requested by reviewer 5EEF. MCNN+Diffusion outperforms all baselines here too.
2. New comparison with vanilla MLP+BC with oversampling of memories in Appendix F (Table 5) for reviewer L3an. MCNN+MLP significantly outperforms MLP+BC with oversampling.
3. Values from Figure 1 have been tabulated in Appendix F (Tables 3 and 4) as requested by reviewer L3an.
4. Added clarification on the Adroit environments in Appendix E for reviewers kyQU and 5EEF.
5. Replaced anonymous code link with anonymous website link (https://sites.google.com/view/mcnn-anon) that also contains videos of the learned policies in Adroit and Franka Kitchen and videos of the human demonstrations in Adroit.

Please let us know if you have lingering questions and whether we can provide any additional clarifications during the discussion period to improve your rating of our paper.

Thanks,

Authors

---

### Meta-Review · Area_Chair_63pv · 2023-12-06

**Metareview:**

The submitted paper proposes the memory-consistent neural network (MCNN), a model class specifically designed for imitation learning to alleviate the compounding error problem. In particular, MCNN's outputs are constrained to be in the vicinity of prototypical training samples. In experiments, the authors demonstrate that the proposed MCNN can outperform vanilla neural networks in imitation learning.

Weaknesses: In contrast to vanilla neural networks, the proposed MCNN is semi-parametric and thus the computational costs increase with the number of memories. However, the authors demonstrated this additional cost is often negligible. The experiments did initially not cover all aspects deemed relevant by the reviewers, e.g., regarding multi-modality, but the authors added additional experiments in this regard.

Strength: The proposed method is simple, well-motivated, and demonstrated to be effective.

All reviewers recommended (weak) acceptance of the paper and I agree with their suggestion. Hence, I recommend acceptance.

**Justification For Why Not Higher Score:**

It's a well-written paper with a relatively easy-to-implement and thoroughly analyzed approach which leads to consistent performance gains but at the same time will not change fundamentally how we approach BC.

**Justification For Why Not Lower Score:**

A decent paper that should be accepted.

---

### Decision · Program_Chairs · 2024-01-16

Accept (poster)